# A brain-wide analysis maps structural evolution to distinct anatomical module

Robert A Kozol[1]*, Andrew J Conith[2], Anders Yuiska[1], Alexia Cree-Newman[1], Bernadeth Tolentino[1], Kasey Benesh[1], Alexandra Paz[1], Evan Lloyd[3], Johanna E Kowalko[4], Alex C Keene[3], Craig Albertson[2], Erik R Duboue[1]*

[1]Jupiter Life Science Initiative, Florida Atlantic University, Jupiter, United States; [2]Department of Biology, University of Massachusetts Amherst, Amherst, United States; [3]Department of Biology, Texas A&M University, College Station, United States; [4]Department of Biological Sciences, Lehigh University, Bethlehem, United States

**Abstract** The vertebrate brain is highly conserved topologically, but less is known about neuroanatomical variation between individual brain regions. Neuroanatomical variation at the regional level is hypothesized to provide functional expansion, building upon ancestral anatomy needed for basic functions. Classically, animal models used to study evolution have lacked tools for detailed anatomical analysis that are widely used in zebrafish and mice, presenting a barrier to studying brain evolution at fine scales. In this study, we sought to investigate the evolution of brain anatomy using a single species of fish consisting of divergent surface and cave morphs, that permits functional genetic testing of regional volume and shape across the entire brain. We generated a high-resolution brain atlas for the blind Mexican cavefish *Astyanax mexicanus* and coupled the atlas with automated computational tools to directly assess variability in brain region shape and volume across all populations. We measured the volume and shape of every grossly defined neuroanatomical region of the brain and assessed correlations between anatomical regions in surface fish, cavefish, and surface × cave $F_2$ hybrids, whose phenotypes span the range of surface to cave. We find that dorsal regions of the brain are contracted, while ventral regions have expanded, with $F_2$ hybrid data providing support for developmental constraint along the dorsal-ventral axis. Furthermore, these dorsal-ventral relationships in anatomical variation show similar patterns for both volume and shape, suggesting that the anatomical evolution captured by these two parameters could be driven by similar developmental mechanisms. Together, these data demonstrate that *A. mexicanus* is a powerful system for functionally determining basic principles of brain evolution and will permit testing how genes influence early patterning events to drive brain-wide anatomical evolution.

**\*For correspondence:**
rkozol@fau.edu (RAK);
eduboue@fau.edu (ERD)

**Competing interest:** The authors declare that no competing interests exist.

## Editor's evaluation

The authors ask if brain regions change based on the functional constraints or developmental constraints. To address this, the authors introduce an automated method for brain segmentation based on the zebrafish tool to study brain evolution in Astyanax.

## Introduction

Regional topology of the brain has remained remarkably conserved in vertebrate lineages across evolution (*Holland and Holland, 2021*; *Charvet et al., 2011*). While the arrangement for subregions of the brain remains constant, the overall size and shape of individual brain regions can vary considerably, even among closely related groups. (*Hoops et al., 2017*; *Pereira-Pedro et al., 2020*;

*Neubauer et al., 2020*; *Briscoe and Ragsdale, 2018*). Comparative neuroanatomical studies suggest that novel anatomical changes are built upon the conservation of ancestral brain anatomy, with anatomical changes resulting in the development of new regions that likely expand function and functional repertoire (*Woych et al., 2022*; *Wolff et al., 2017*). Furthermore, the convergence of similar functional elaborations of the vertebrate brain, despite independent lineages of evolution, suggest some regions of the brain are primed for anatomical and ultimately functional diversification (*Schumacher and Carlson, 2022*; *Emery and Clayton, 2004*; *Northcutt, 2002*). Although these comparative studies have provided a wealth of insight into the evolutionary history of neuroanatomy, pioneers in the field strongly advise diversifying our animal model pool to permit functional tests to theories underlying the evolution of the vertebrate brain (*Northcutt, 2002*; *Striedter, 1998*).

Two central hypotheses are thought to drive anatomical brain evolution; the first suggesting that subregions brain-wide tend to evolve together, with selection operating on mechanisms that govern the growth of all regions, and the second positing that selection can act on individual brain regions, and that regions which are functionally related will anatomically evolve together independent of other brain regions (*Barton and Harvey, 2000*; *Herculano-Houzel et al., 2014*; *Montgomery et al., 2016*). While data supporting each hypothesis exist, the large divergence times and poor understanding of evolutionary history in most comparative models makes generalizing these theories difficult. Additionally, these theories have largely been applied to studies analyzing anatomical size (*Barton and Harvey, 2000*; *Wartel et al., 2019*; *Axelrod et al., 2018*), overlooking how 3D shape of brain regions are impacted by evolutionary processes. Moreover, the relationship between the evolution of the size and shape of distinct anatomical regions is poorly understood, and it is unclear how these two important aspects of neuroanatomy explain the evolution of the brain.

While volume and shape are known to govern anatomical variation across the brain, there is still uncertainty of whether similar or distinct mechanisms govern both parameters (*Reardon et al., 2018*). However, most comparative studies tend to focus on either volume or shape, with some volume to shape analyses comparing trends across independent studies (*Gómez-Robles et al., 2014*; *Sansalone et al., 2020*; *Montgomery et al., 2021*). Current models to explore mechanisms driving volume and shape rely on non-model systems that lack experimental approaches, or model organisms that lack genetic diversity, creating an impediment for investigating basic principles of brain evolution. (*Ponce de Leon et al., 2021*; *Mitchell, 1977*). Non-traditional models can bridge this experimental gap, including experimental approaches to determine whether similar genetic and developmental mechanisms underlie general principles of evolution.

The blind Mexican cavefish *Astyanax mexicanus* provides a powerful model for directly testing how genetic variation impacts brain-wide anatomical evolution (*Mitchell, 1977*; *Jeffery, 2008*). *A. mexicanus* exists as a species with two distinct forms: river dwelling surface fish and cave dwelling populations that have independently evolved troglobitic phenotypes (*Bradic et al., 2012*; *Gross, 2012*). This separation has led to high genetic diversity between populations which underlies the stark differences in phenotypes between surface and cave populations (*Borowsky, 2021*; *Warren et al., 2021*). Importantly, surface × cave hybrid offspring are biologically viable, allowing us to exploit the genetic differences between each population, and ultimately identify the genetic underpinnings of neuroanatomical evolution in the cavefish brain (*O'Gorman et al., 2021*; *Duboué et al., 2011*). Therefore, a hybrid population analysis using novel neurocomputational tools can be used to study covariation of neuroanatomy across a well-annotated atlas, and directly test whether cavefish brains exhibit support for either the developmental or functional constraint hypothesis. Finally, the relationship between brain region shape and volume can be analyzed in comparative and direct analyses, which will be critical in understanding how brain regions evolve in relationship to one another.

In the current study, we generated a brain-wide neuroanatomical atlas for *A. mexicanus* and applied new computational tools for assessing brain-wide changes in both brain region volume (*Gupta et al., 2018*) and shape (*Conith et al., 2019*). We then applied this atlas to hybrid brains to make associations between naturally occurring genetic variation of wildtype populations and neuroanatomical phenotypes. Our surface × cave hybrid data reveal that brain-region volume and shape are genetically specified to regulate brain-wide anatomical evolution in *A. mexicanus*. Furthermore, volume and shape exhibit similarities in brain-wide anatomical covariation, suggesting that these two parameters share developmental mechanisms that are causing cavefish brains to contract dorsally and expand

ventrally. These results suggest that selection may be operating on simple developmental mechanisms, that likely impact early patterning events to modulate the volume and shape of brain regions.

## Results

### Generation of a single brain-wide atlas for all *Astyanax* morphs

To analyze regional variation in brain anatomy, we created a single atlas for all *A. mexicanus* morphs to provide neuroanatomical comparisons across surface, cave, and surface × cave hybrid populations (*Figure 1a–c*). A neuroanatomical analysis pipeline from zebrafish that performs automated segmentation of brains was then adapted and tested on *A. mexicanus* brains (*Gupta et al., 2018*; *Figure 1b and c* and *Figure 1—figure supplement 1a–c*). This tool provides a single atlas that can be continually segmented through brain regions to identify neuroanatomical differences across various molecularly and functionally defined sub-nuclei (*Figure 1c*, *Figure 1—figure supplement 1d, e*). The segmentation accuracy was confirmed by a pairwise cross correlation of tERK staining and manual to automated segmentation overlap (*Figure 1—figure supplement 2a, b*; >98% tERK cross-correlation and >78% segmentation cross-correlation). Previous brain atlases for model organisms were constructed using molecular markers that are known to demarcate specific subregions, such as transgenic lines and antibody labeling (*Gupta et al., 2018*; *Randlett et al., 2015*; *Kunst et al., 2019*). To determine whether our atlas maintains similar molecular accuracy, we developed a dual staining technique that combines RNA hybridization chain reaction (HCR) in situ hybridization with immunohistochemistry (IHC), resulting in automated segmentation of RNA in situ probes via total-ERK antibody registration. With this approach, we were able to confirm the accuracy of larger segments identified through automated segmentation, including subregions of the hypothalamus and optic tectum, with accurate segment bounding of *insulin gene enhancer protein (ISL-1)* (*Figure 1d*, *Randlett et al., 2015*; *Kunst et al., 2019*; *Sanek and Grinblat, 2008*; *Langenberg and Brand, 2005*) and *orthodenticle homeobox 2* (*otx2*) RNA labeling, respectively (*Figure 1—figure supplement 3a*, *French et al., 2007*; *Paridaen et al., 2009*; *Diotel et al., 2015*). We then tested the accuracy of smaller regions, such as the dorsal subpallium, medial preoptic region, and thalamus, via *gastrulation brain homeobox 1* (*gbx1*), *oxytocin* (*oxt*) and *nitrous oxide 1* (*nos1*) RNA labeling, respectively (*Figure 1—figure supplement 3b–d*, *Randlett et al., 2015*; *Kunst et al., 2019*; *Blechman et al., 2007*; *Gutierrez-Triana et al., 2014*). Finally, we confirmed the accuracy of the smallest subregions of the brain that can be defined molecularly, such as the locus coeruleus and dorsal raphe, using tyrosine hydroxylase (TH) and 5-hydroxytryptamine (5-HT) antibody labeling, respectively (*Figure 1—figure supplement 3e,f*, *Gupta et al., 2018*; *Randlett et al., 2015*; *Sittaramane et al., 2009*; *Kidwell et al., 2018*; *Oikonomou et al., 2019*; *Ulhaq and Kishida, 2018*).

### Determining neuroanatomical variation brain-wide for surface and cave populations

To analyze and compare the relative volume of brain regions in surface fish and cavefish populations, volumetric data was measured and analyzed for variation between surface and cave brains (*Figure 2*, *Figure 1—figure supplement 1e*). The atlas was applied to individuals from the Pachón cavefish, Molino cavefish, Rio Choy surface fish, Pachón to Rio Choy $F_1$ and $F_2$ hybrid, and Molino to Rio Choy $F_2$ hybrid populations, via immunolabel-based brain registration and inverse registration, allowing us to address the evolutionary mechanisms underlying variation in brain anatomy. Importantly, Pachón and Molino cavefish are independently evolved populations (*Herman et al., 2018*), that allow us to determine whether the process of evolution impacts neuroanatomy convergently in cave environments, despite differences in the standing genetic variation of the two populations.

To survey volumetric variation across populations, we progressively segmented subdivisions within our brain atlas from larger gross anatomical segments to molecularly defined subregions. This progressive segmentation provided an analytical tool for defining regional variability through sub-nuclei, with localization of variability increasing as we scaled through each level of the atlas (*Figure 2a–c*, *Figure 2—figure supplement 1*). Our initial analysis of numerous brain regions revealed results that support previously published studies for the four major brain regions and larger subdivisions of those brain regions (*Figure 2—figure supplement 1a, b*; *Loomis et al., 2019*; *Jaggard et al., 2020*). While this comparison found volumetric differences that were previously

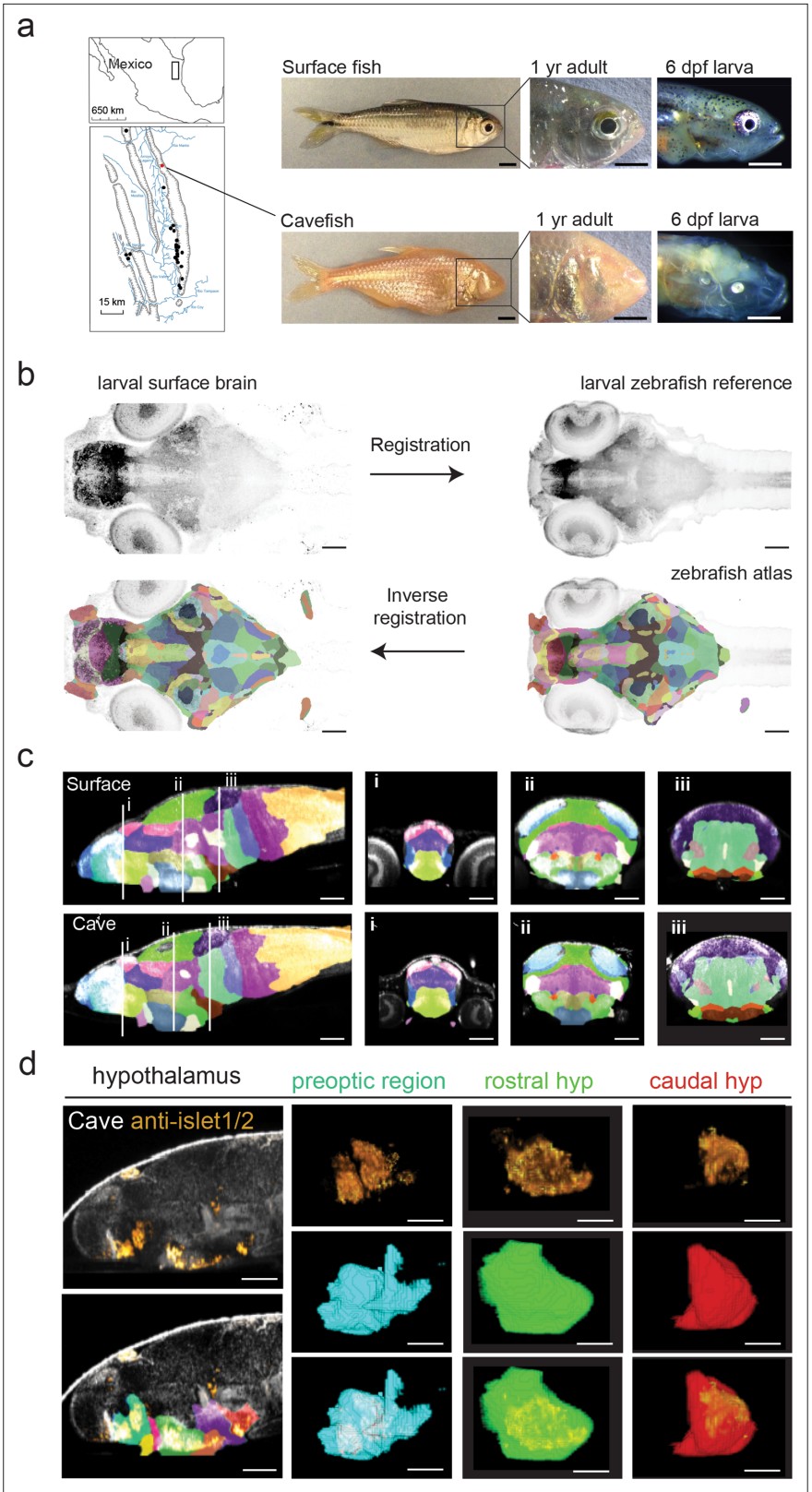

**Figure 1.** Developing a single *A. mexicanus* atlas to perform direct brain-wide morphometric analyses across all populations. (**a**) Map showing the 29 independently evolved cave populations (black dots) of the El Abra region in Mexico. The Pachón cavefish population used for this project is marked as a red dot. Scale bars = 0.5 cm (full fish, 1 year adult) and 0.5 mm (larvae). (**b**) Schematic showing registration and atlas inverse registration

*Figure 1 continued on next page*

*Figure 1 continued*

method used to create an *A. mexicanus* atlas for cross-population segmentation and analysis. (**c**) Sagittal and transverse (i–iii) sections of the 26 region surface fish and cavefish atlas. (**i**) Habenula (pink), pallium (blue), ventral thalamus (purple), and preoptic (light green). (**ii**) optic tectum neuropil (sky blue), optic tectum cell bodies (green), tegmentum (light purple), rostral hypothalamus (dark blue), posterior tuberculum (gold), statoacoustic ganglion (beige). (**iii**) Cerebellum (dark purple), prepontine (light green), locus coeruleus (brown), raphe (beige), intermediate hypothalamus (dark brown), and caudal hypothalamus (bright red). (**d**) Islet1/2 antibody segmentation following ANTs inverse registration of cavefish atlas. Islet positive neurons exhibit the same segmentation in the preoptic, rostral, and caudal portions of the hypothalamus that have been reported islet positive in zebrafish. Scale bars (**b–d**) = 80 μm.

The online version of this article includes the following source data and figure supplement(s) for figure 1:

**Source data 1.** Surface × Pachón $F_2$ hybrid brain atlas in nifty format.

**Figure supplement 1.** Pipeline for immunohistochemistry, automated segmentation, and volumetric comparisons of individual fish larvae.

**Figure supplement 2.** Cross-correlation analysis between hand and automated segmentation of total-ERK-defined brain segments.

**Figure supplement 3.** RNA probe and antibody analysis of segmentation accuracy across the *Astyanax* atlas.

reported in broad developmental regions (*Jaggard et al., 2020*; *Menuet et al., 2007*), such as the hypothalamus (*Figure 2b*; F=9.252, surface to Pachón p=0.0154, surface to Molino p=0.003), our atlas was able to determine that the intermediate and caudal hypothalamus were enlarged in cavefish populations (*Figure 2c*, *Figure 2—figure supplement 1b*; intermediate – F=19.11, surface to Pachón p<0.0001, surface to Molino p<0.0001; caudal – F=10.98, surface to Pachón p<0.0001, surface to Molino p<0.0001). In addition, we also discovered novel volumetric differences, including contraction of the dorsal diencephalon in cavefish (*Figure 2b*; F=39.89, surface to Pachón p=0.0025, surface to Molino p<0.0001), that we localized to the dorsal thalamus in Pachón cavefish and across the thalamus and habenula in Molino cavefish (*Figure 2c*, dorsal thalamus – F=16.64, surface to Pachón p<0.0001, surface to Molino p<0.0001; Habenula – F=16.64, surface to Molino p<0.0001). Overall, we were able to use this single atlas to pinpoint discrete differences between brain regions of surface fish and cavefish, while also creating a brain-wide model for Pachón and Molino cavefish that highlights a convergent dorsal-ventral remodeling of the brain in both populations (*Figure 2d*).

## Analysis of hybrid animals defines neuroanatomical associations brain-wide

Lab-generated hybridization between surface and cave populations provides a powerful system for determining how high genetic diversity of natural populations contributes to phenotypic diversity. To define anatomical relationships volumetrically between wildtype populations and larval offspring, we quantified relative volume for each brain region of surface, cave, and surface × cave $F_1$ and $F_2$ hybrids (*Figure 3*, *Figure 3—figure supplement 1*). Hybrid brain regions show variability that appears consistent with different modes of inheritance, including surface dominant, cavefish dominant, and surface × cavefish intermediate anatomical forms (*Figure 3—figure supplement 1a–c*). We then investigated regional variability in surface × cave $F_2$ hybrid brain regions by further segmenting down to local sub-nuclei (*Figure 3a–d*, *Figure 3—figure supplement 2a–c*). These analyses revealed molecularly defined regions that account for segment variability, such as the ventral sub-nuclei of the optic tectum stratum periventricular (*Figure 3d*; ventral optic tectum stratum periventricular – F=34.61, surface to surface × Pachón $F_2$ p<0.0001, Pachón to surface × Pachón $F_2$ p<0.0001, surface to surface × Molino $F_2$ p<0.0001, and Molino to surface × Molino $F_2$ p=0.0005) and pallium (*Figure 3—figure supplement 2c*, ventral pallium – F=43.56, surface to surface × Pachón $F_2$ p<0.0438, Pachón to surface × Pachón $F_2$ p<0.0001, surface to surface × Molino $F_2$ p<0.0001, and Molino to surface × Molino $F_2$ p=0.0002). These results reveal that brain-wide anatomical variation is likely genetically heritable in cavefish and that this genetic relationship can be resolved at the sub-nuclei level across the brain.

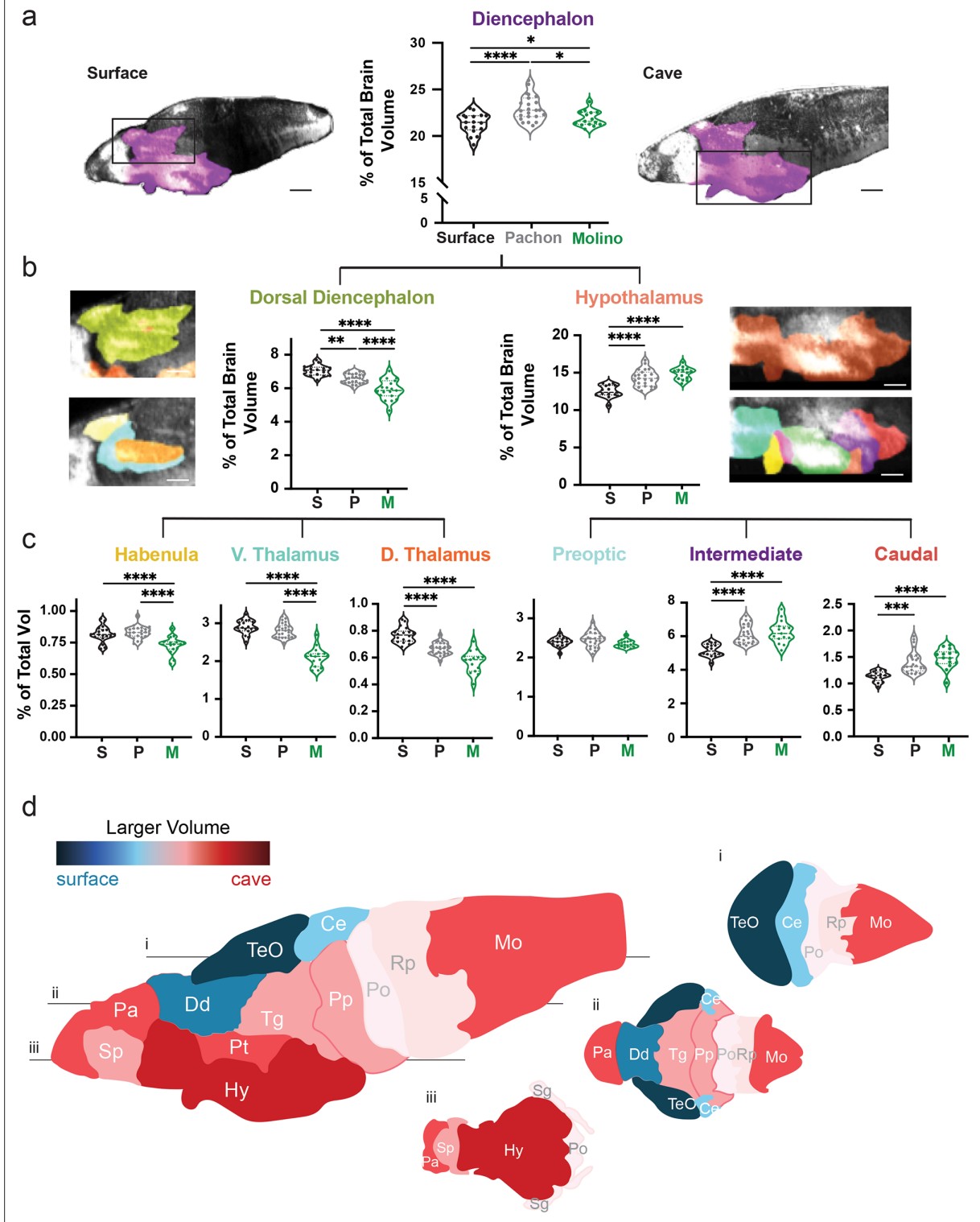

**Figure 2.** Volumetric variation in wildtype populations reveal convergent dorsal contraction and ventral expansion in the brain of two cavefish populations. (**a**) Volumetric comparison of the diencephalon in surface fish, Pachón and Molino cavefish. Percent total brain volume represents pixels of segment divided by total pixels in the brain. Sagittal sections show diencephalon (purple). (**b**) Volumetric comparisons of the dorsal diencephalon (green) and hypothalamus (orange). (**c**) Volumetric comparisons of the habenula (gold), ventral thalamus (teal) and dorsal thalamus (burnt orange) of the dorsal diencephalon; along with the preoptic (cyan), intermediate zone (purple), and caudal zone (red), of the hypothalamus. Sample size = surface (16) and Pachón cavefish (24). (**d**) Colorimetric model depicting size differences in brain regions between surface fish and cavefish. A larger volume in surface fish results in blue coloration, while a larger volume in cavefish results in a red coloration. Horizontal optical sections depicting (**i**) dorsal, (**ii**) medial, and

*Figure 2 continued on next page*

*Figure 2 continued*

(iii) ventral views of the brain. p-Value significance is coded as: *=p < 0.05, **=p < 0.01, ***=p < 0.001, ****=p < 0.0001. Statistical tables can be found in the Dryad repository associated with this study (**Portella et al., 2010**). Scale bars = 80 μm (**a**), 25 μm (**b**).

The online version of this article includes the following source data and figure supplement(s) for figure 2:

**Source data 1.** Volumetric values for the diencephalon of wildtype larvae.

**Source data 2.** Volumetric values for the dorsal diencephalon of wilid-type larvae.

**Source data 3.** Volumetric values for the habenula of wildtype larvae.

**Source data 4.** Volumetric values for the ventral thalamus of wildtype larvae.

**Source data 5.** Volumetric values for the dorsal thalamus of wildtype larvae.

**Source data 6.** Volumetric values for the hypothalamus of wildtype larvae.

**Source data 7.** Volumetric values for the preoptic region of wildtype larvae.

**Source data 8.** Volumetric values for the intermediate zone of wildtype individual larvae.

**Source data 9.** Volumetric values for the caudal hypothalamus of wildtype individual larvae.

**Figure supplement 1.** Variation in segment volume between surface and cavefish populations.

## Covariation of $F_2$ hybrid brain regions reveals brain-wide anatomical tradeoff impacting distinct developmental clusters

To determine which brain regions covary and whether anatomical variation provides support for either the developmental or functional constraint hypothesis, we looked for pairwise anatomical associations for regional volume across all subregions of the brain in surface to Pachón and surface to Molino $F_2$ hybrids. These results were then run through a hierarchical cluster analysis to gain insight into brain-wide evolutionary mechanisms driving anatomical change in cavefish brains. For this dataset, clusters constitute regions that share volumetric associations, wherein $F_2$ brain regions are either getting larger or smaller together (*Figure 4a–d*). The clustering analysis for the 180-brain region scale revealed six large clusters for surface to Pachón $F_2$ hybrids (*Figure 4b*) and twelve clusters for surface to Molino $F_2$ hybrids (*Figure 4d*), with each cluster showing strong positive volumetric associations among subregions in that cluster. We also found strong negative correlations between cluster groups (*Figure 4b and d*), suggesting that these regions have the potential to co-evolve by similar genetic mechanisms, with one group getting larger as the other gets smaller. Surprisingly, these clusters map onto the brain in well-defined dorsal to ventral positions, with positive associations being found across dorsal clusters and ventral clusters, and negative associations between dorsally and ventrally positioned clusters (*Figure 4e, f*). This first analysis suggests that small subregions of the brain are clustering as larger modules and exhibiting brain-wide volumetric associations that suggest an anatomical tradeoff along the dorsal-ventral axis, where some areas become reduced in size at the expense of other areas' increasing volumes.

To help map these brain-wide volumetric associations to larger developmental regions, we reduced our segmentation to 13 ontologically defined regions (e.g. hypothalamus, cerebellum, etc.), we then performed pairwise correlation and cluster analyses on our 13 brain region scale atlas for the two populations of surface × cave $F_2$ hybrids (*Figure 4—figure supplement 1a–d*). This developmental cluster analysis revealed three clusters for both populations (*Figure 4—figure supplement 1c, d*), with positive associations between neuroanatomical areas within a cluster. We then mapped neuro-anatomical regions with the clusters back on the brain and found that loci within each cluster were physically localized together (*Figure 4—figure supplement 1e*). The large clusters for both populations encompassed the same broad anatomical regions, with one cluster comprised of the dorsal and caudal areas of the brain (e.g. optic tectum and cerebellum), while the second cluster was predominantly made up of the ventral brain (e.g. hypothalamus and subpallium; *Figure 4—figure supplement 1e, f*). When compared to surface to Pachón hybrids, only the statoacoustic ganglion clustered differently in surface to Molino hybrids (*Figure 4—figure supplement 1c, d*). Therefore, we found that the two large portions of the brain exhibit the same dorsal ventral volumetric associations as the smaller clusters found in our 180-brain region analysis. To further analyze the data statistically, we added up the correlation values of the clusters and ran a pairwise comparison across clusters (clusters 1 and 2, t=18.48, p<0.0001; clusters 1 and 3, t=13.82, p<0.0001; clusters 2 and 3, t=5.802, p=0.0011) that revealed statistical significance across all clusters displaying negative volume associations. Taken

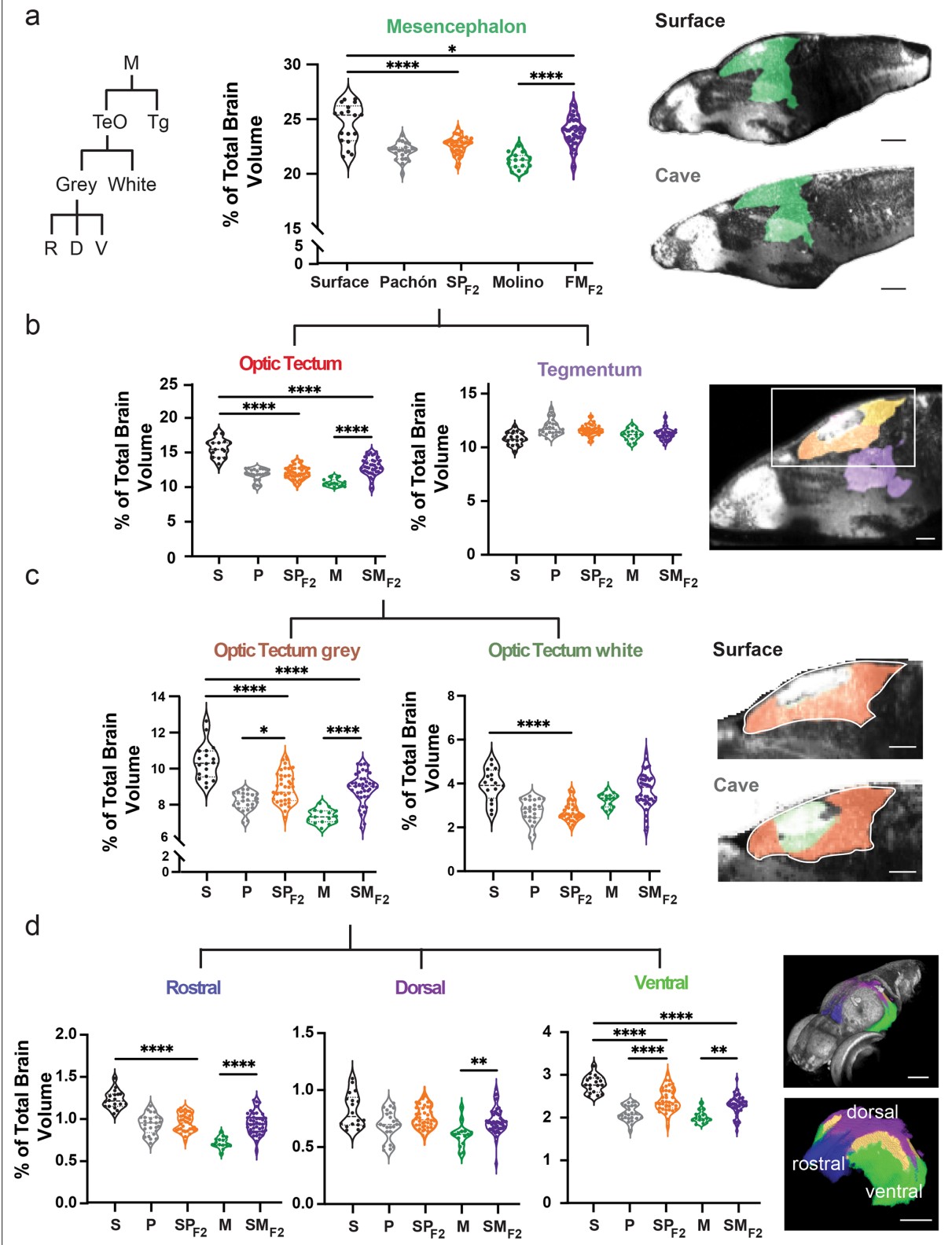

**Figure 3.** Scalable segmentation of the tectum identifies high variability in the ventral sub-nuclei of the optic tectum's cell layers. (**a**) Volumetric comparison of the mesencephalon in surface fish, Pachón cavefish, Molino cavefish, surface × Pachón $F_2$ hybrid ($SPF_2$), and surface × Molino $F_2$ hybrid ($SMF_2$) larvae. Sagittal sections showing the mesencephalon (green). Percent total brain volume represents pixels of segment divided by total pixels in the brain. Segment tree abbreviations, M – mesencephalon, TeO – optic tectum, Tg – tegmentum, R – rostral, D – dorsal, V – ventral. (**b**) Volumetric

*Figure 3 continued on next page*

*Figure 3 continued*

comparisons of the optic tectum (yellow) and tegmentum (purple). (**c**) Volumetric comparisons of the optic tectum white (neuropil; forest green) and gray matter (cell bodies; orange). (**d**) Volumetric comparisons of rostral (royal blue), dorsal (purple), and ventral (lime green) segments of the optic tectum gray matter. All segments were statistically analyzed using a standard ANOVA and Holm's corrected for multiple comparisons. p-Value significance is coded as: *=p < 0.05, **=p < 0.01, ***=p < 0.001, ****=p < 0.0001. Statistical tables can be found in the Dryad repository associated with this study (*Portella et al., 2010*). Scale bars = 80 µm (**a**), 25 µm (**b**), 50 µm (**c and d**).

The online version of this article includes the following source data and figure supplement(s) for figure 3:

**Source data 1.** Volumetric values for the mesencephalon of wildtype and $F_2$ hybrid larvae.

**Source data 2.** Volumetric values for the tegmentum of wildtype and $F_2$ hybrid larvae.

**Source data 3.** Volumetric values for the optic tectum of wildtype and $F_2$ hybrid larvae.

**Source data 4.** Volumetric values for the optic tectum gray matter of wildtype and $F_2$ hybrid larvae.

**Source data 5.** Volumetric values for the optic tectum white matter of wildtype and $F_2$ hybrid larvae.

**Source data 6.** Volumetric values for the rostral optic tectum of wildtype and $F_2$ hybrid larvae.

**Source data 7.** Volumetric values for the dorsal optic tectum of wildtype and $F_2$ hybrid larvae.

**Source data 8.** Volumetric values for the ventral optic tectum of wildtype and $F_2$ hybrid larvae.

**Figure supplement 1.** Volumetric variability in hybrid larvae reflects wildtype genetic diversity through dominant and intermediate phenotypes.

**Figure supplement 2.** Hybrid brains link genetic variation in wildtype populations to anatomical variation in distinct sub-nuclei of the olfactory bulb, subpallium, and pallium.

together, these analyses suggest that a general feature of neuroanatomical evolution in cave-derived populations of *A. mexicanus* may be a developmental tradeoff between ventral expansion and dorsal contraction, as evinced by parallel findings in two independently evolved populations.

## Geometric morphometrics provide an analytical tool for understanding the relationship between shape and volume during brain-wide evolution

Previous studies examining variation in the brain have mostly focused on volume (*Hoops et al., 2017*; *Eliason et al., 2021*) or shape (*Pereira-Pedro et al., 2020*; *Watanabe et al., 2021*), with few providing a comparison of how shape and volume vary brain-wide (*Axelrod et al., 2018*; *Reardon et al., 2018*; *Sansalone et al., 2020*). We sought to examine whether shape variation follows similar patterns as volume, which could suggest shared genetic or developmental origins underlying variation, or whether shape and volume were unrelated. To determine morphological variation in shape across the brain, we employed shape analysis (i.e. geometric morphometrics) approaches previously used in assessing shape variation among whole brain and brain regions (*Conith et al., 2020*; *Conith and Albertson, 2021*). We first examined whether shape showed variation between populations for regions with no variation in volume, then how volume and shape relate within specific regions and finally whether shape variations follow the same brain-wide patterns seen in volumetric variation.

To begin evaluating how subregion shape varies between surface fish and cavefish brains, we chose to characterize the pineal and preoptic region because they show no volumetric variation across populations, yet play functional roles in behaviors that are highly variable across *Astyanax* populations. We characterized pineal and preoptic shape among the three populations using landmark-based geometric morphometrics. We performed a principal component analysis (PCA) to reduce our landmark data to a series of orthogonal axes that best represent shape variation within our brain regions. We found significant differences in shape of the pineal and preoptic region among wildtype surface fish, cavefish, and surface × cave $F_2$ hybrids (*Figure 5—figure supplements 1 and 2*). Importantly, surface × cave hybrids have a range of phenotypes that likely exhibit additive (preoptic, *Figure 5—figure supplement 1a*, F=5.076, Pr(>F)=0.001) and genetically dominant (pineal, *Figure 5—figure supplement 2a*, F=4.159, Pr(>F)=0.0001) modes of inheritance, suggesting that the differences in shape may be driven by genetics. Taken together, we find that despite a lack of volumetric differences among populations, population-level variation in regional brain shape can occur, and the specific variation we observed likely impacts adaptive behaviors discovered in previous studies.

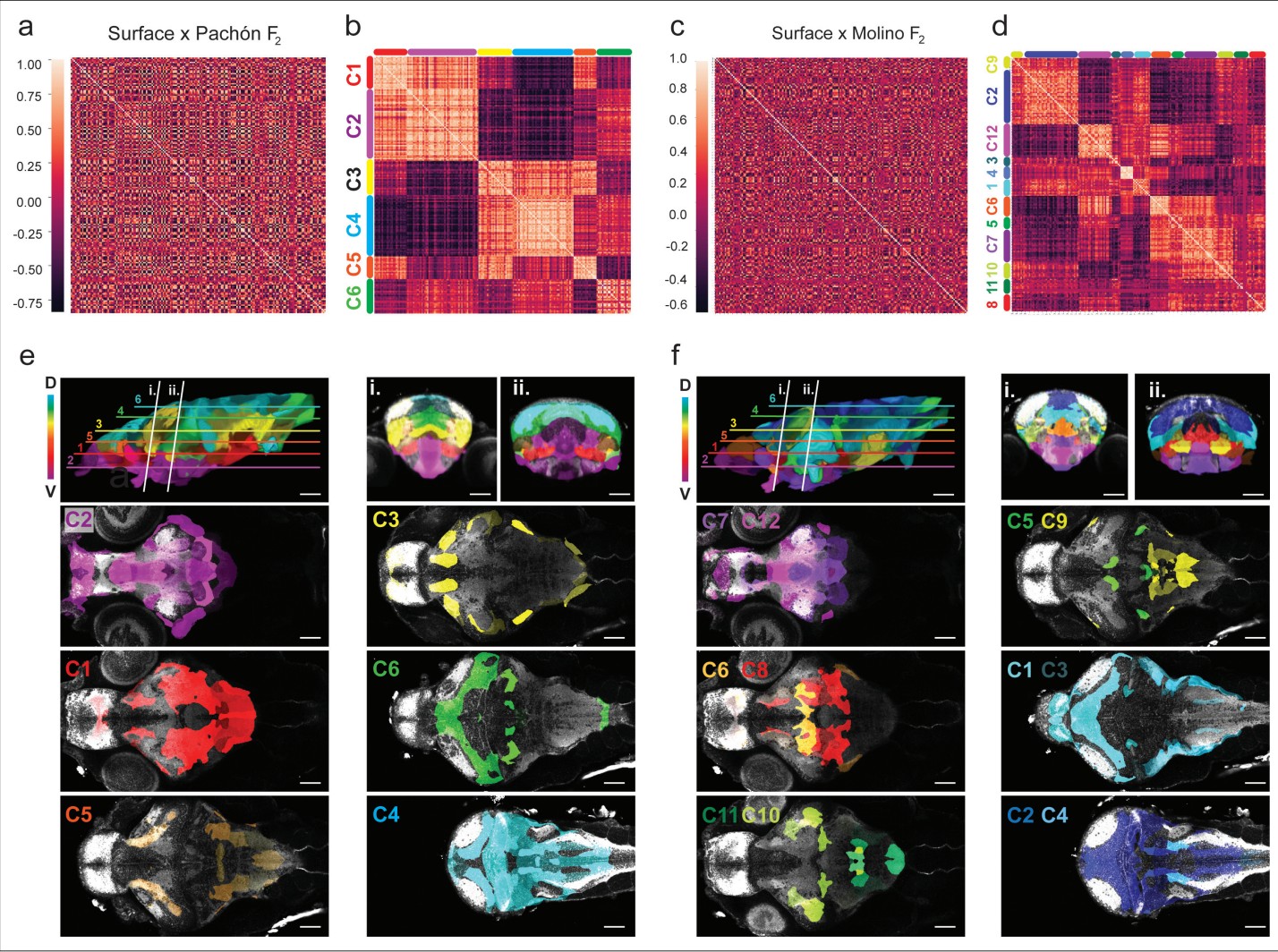

**Figure 4.** Volumetric covariation and clustering of hybrid brain regions reveals convergent associations across the dorsal-ventral axis. (**a**) Cross-correlation analysis of surface to Pachón F₂ hybrids for the 180 segmented *Astyanax* brain atlas. (**b**) Cluster analysis array showing six clusters exhibiting positive volumetric associations. Positive relationships are color-coded light red, negative dark red (n=37). (**c**) Cross-correlation analysis of surface to Molino F₂ hybrids for the 180 segmented *Astyanax* brain atlas. (**b**) Cluster analysis array showing 12 clusters exhibiting positive volumetric associations a. Positive associations are color-coded light red, negative dark red (n=37). Clusters color-coded and mapped onto the surface × cave F₂ hybrid reference brain for both (**e**) surface to Pachón F₂ hybrids and (**f**) surface to Molino F2 hybrids. The rainbow gradient represents depth along the z-plane, blue shifted (dorsal) to red shifted (ventral). Statistical tables can be found in the Dryad repository associated with this study (*Portella et al., 2010*).

The online version of this article includes the following source data and figure supplement(s) for figure 4:

**Source data 1.** Correlation coefficient matrix for 180-brain region atlas volumetric comparisons of surface × Pachón F₂ hybrid larvae.

**Source data 2.** Correlation coefficient matrix for 180-brain region atlas volumetric covariation of surface × Molino F₂ hybrid larvae.

**Figure supplement 1.** Covariation of brain region size reveals developmental tradeoff between dorsal-ventral clusters brain-wide in independently derived cavefish populations.

## Shape and volume relationships within regions can exhibit similarities or variation on a region-to-region basis

To determine whether features of shape and volume of regions covary in relation to brain-wide anatomical evolution in cavefish, we chose to analyze two regions from our volumetric cluster 1, the optic tectum and cerebellum, and two regions from cluster 2, the hypothalamus and tegmentum (*Figure 5a, b*). These regions allowed us to test whether shape exhibits covariation patterns similar to volume, including positive relationships within and negative relationships across dorsal and ventral clusters. First, we analyzed shape variation across F₂ individuals for each brain region. To that end,

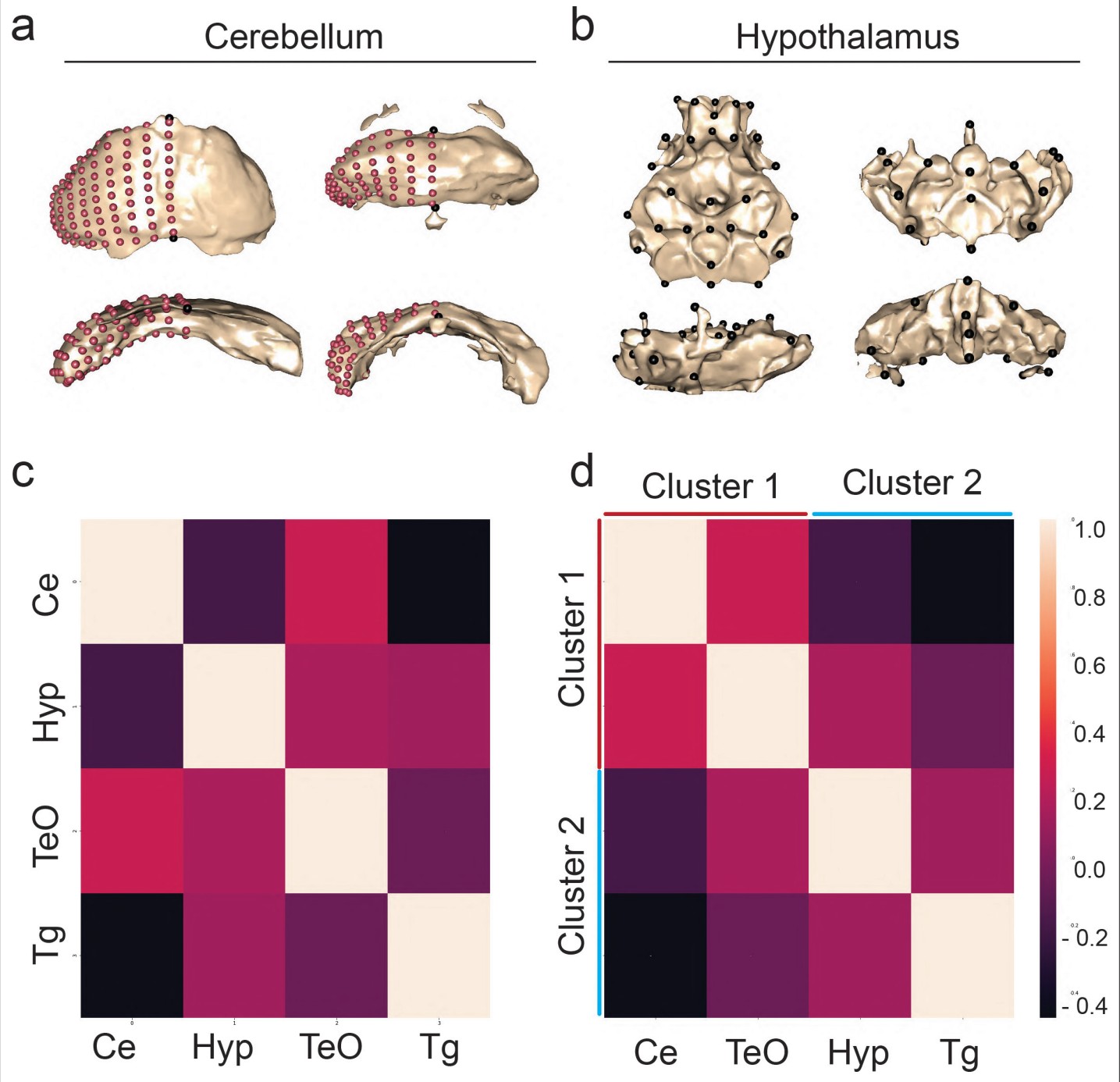

**Figure 5.** Shape covariation suggests volume and shape share brain-wide mechanism of brain evolution. Representatives of shape for the cerebellum and hypothalamus of pure populations (**a**) principal component 1 (PC1) and (**b**) principal component 2 (PC2). (**c**) Correlation matrix comparing the covariation of shape between regions from volumetric covarying cluster 1, cerebellum (Ce) and optic tectum (TeO), and cluster 2, hypothalamus (Hyp) and tegmentum (Tg). Sample size, n=37. (**d**) A cluster analysis of covariation grouped regions into two clusters as predicted by volumetric covariation. Scale bars = 100 μm. Statistical tables can be found in the Dryad repository associated with this study (***Portella et al., 2010***).

The online version of this article includes the following source data and figure supplement(s) for figure 5:

**Source data 1.** Correlation coefficient matrix for 13 brain region atlas shape covariation of surface × Pachón $F_2$ hybrid larvae.

**Figure supplement 1.** Shape variability of the preoptic region in hybrid larvae display an intermediate phenotype between wildtype populations.

**Figure supplement 2.** Pineal shape variation in hybrids exhibits a cavefish dominant phenotype.

**Figure supplement 3.** Optic tectum shape variation is characterized by width curvature and thickness.

*Figure 5 continued on next page*

*Figure 5 continued*

**Figure supplement 4.** Cerebellar shape variation is characterized by depth of curvature and thickness.

**Figure supplement 5.** Hypothalamic shape variation is characterized by length, central thickness, and posterior width.

**Figure supplement 6.** Tegmentum shape variation is characterized by depth and length.

we again performed a PCA to characterize shape variation and identify what aspects of shape were driving differences within the $F_2$ hybrid population for each of the four brain regions (*Figure 5a*, *Figure 5—figure supplements 3–6*).

We then assessed the degree of association among these four brain regions using partial least squares (PLS), a method which permitted the complete landmark configuration of each individual to be used in assessing the degree of covariation. We found that our shape × shape associations broadly match patterns observed in the volumetric data, as we observed significant covariation in shape between the optic tectum and cerebellum (Z=2.48, p=0.004) alongside covariation between the hypothalamus and tegmentum (Z=4.026, p=0.0001). However, we also observed covariation in shape between the hypothalamus and cerebellum (Z=3.146, p=0.0001), indicating that the regulation of shape and volume is likely under distinct development control.

Given that certain regions of the brain could exhibit differences in shape among populations (i.e. preoptic, pineal) independent of volume, we directly assessed the degree of association between shape and volume for each brain region. We found significant associations between volume and shape in three of four brain regions. There were strong associations between volume and shape in the cerebellum, tectum, and tegmentum, while we found no association in the hypothalamus (tegmentum, Z=2.686, Pr(>F)=0.0041; optic tectum, Z=2.06, Pr(>F)=0.0178; cerebellum, Z=2.752, Pr(>F)=0.0021). These results show that the relationship between volume and shape varies among individual brain regions, suggesting that shape and volume can exhibit either shared or distinct mechanisms.

## Shape and volume variation follow the same covariation pattern brain-wide suggesting shared developmental mechanisms of brain evolution

We sought to compare regional shape variation across the brain to better understand the similarities and differences between how shape and volume evolve in the brain. To determine whether shape and volume were modulated by distinct or similar mechanisms, we extracted the first principal component from our optic tectum, cerebellum, hypothalamus, and tegmentum (*Figure 5a, b*). By extracting a single variable we could assess associations among the shapes of brain regions using the same cluster-based methodological approaches that were applied to the volumetric data. As a result, we could determine whether variation in shape and volume are modified by distinct or varying developmental mechanisms. We found that covariation of anatomical shape clusters the same as volume in a dorsal-ventral fashion, with cluster 1 and cluster 2 showing positive relationships within clusters, and negative relationships across clusters (*Figure 5c, d*). This initial shape analysis suggests that mechanisms in support of the developmental constraint hypothesis may be impacting both anatomical volume and shape to reorganize the dorsal-ventral development of cavefish brains. However, future efforts looking at all brain regions will be needed for a stronger conclusion.

## Discussion

Here, we establish a laboratory model of anatomical brain evolution that utilizes an innovative, molecularly defined neuroanatomical atlas and applied computational tools, which can be used to assess mechanisms underlying anatomical brain evolution. The application of this atlas and these computational approaches to $F_2$ hybrid fish permits a brain-wide dissection of how neuroanatomy changes, and a powerful analysis of not only how different neuroanatomical areas evolve but also which areas co-segregate together. These studies suggest that brains of cave-adapted populations of *A. mexicanus* are anatomically evolving via the developmental constraint hypothesis (*Herculano-Houzel et al., 2014*; *Montgomery et al., 2016*), with a brain-wide dorsal-ventral relationship, that suggests expansion of ventral regions is directly related to contraction of dorsal regions. Finally, this study is one of the first to directly assess how the volume and shape of brain regions relate to one another across genetically and phenotypically diverse populations of a single species.

Previous studies examining how the brain evolves have largely been restricted to comparative analyses between closely related, albeit different species, and these studies have revealed gross differences in neuroanatomy, connectivity, and function between derived animals (*Reardon et al., 2018*; *Gómez-Robles et al., 2014*; *Sansalone et al., 2020*; *Montgomery et al., 2021*). However, the *A. mexicanus* model system provides a powerful tool for assessing how the brain evolves in a single species with multiple divergent forms and an extant ancestor (*Jeffery, 2008*; *Gross, 2012*; *Jeffery, 2001*). Moreover, because surface and cave forms are the same species, the ability to produce surface/cave and cave/cave $F_2$ hybrid fish permits a powerful dissection of functional principles underlying brain evolution (*O'Gorman et al., 2021*; *Duboué et al., 2011*). We previously published population-specific neuroanatomical atlases for this species and used these to examine how gross neuroanatomy differs between surface and cave fish, and how physiology relates to behavior (*Loomis et al., 2019*; *Jaggard et al., 2020*). The current study extends applications of this model to further understand how the brain evolves, and includes a single atlas for all populations to functionally compare neuroanatomy in pure and hybrid offspring, automated brain segmentation for 180 annotated sub-populations of neurons, and the application of computational approaches for a complete whole-brain assessment of the evolution of the brain. In future studies, we will be able to utilize the genetic diversity of these populations to map anatomical traits and then functionally test how natural genetic variation in parental populations (e.g. surface, Pachón, etc.) impact the anatomical evolution of the cavefish brain.

Two competing hypotheses exist to explain brain evolution: one theory suggests that the majority of the brain evolves anatomically via changes to shared developmental programs, whereas others have suggested that more discrete regions will independently evolve based on shared function (*Herculano-Houzel et al., 2014*; *Montgomery et al., 2016*). Our data from two independent cavefish populations provides evidence that supports the notion that the developmentally related regions of the brain co-evolve, with the dorsal-caudal areas of the brain evolving together, and that regions such as the optic tectum and the cerebellum, two areas that constitute a large proportion of the dorsal-caudal region shrink in size. In contrast, rostral-ventral areas co-evolve together, such as the hypothalamus and subpallium, that are enlarged in cavefish. Importantly, we find in $F_2$ hybrid fish that reduced optic tectum and cerebellum are concomitant with an enlarged hypothalamus and subpallium, suggesting that expansion of some regions come at the expense of others. This anatomical outcome may suggest that early brain patterning genes and developmental mechanisms could be influencing the establishment of the dorsal-ventral axis of the brain (*O'Gorman et al., 2021*; *Duboué et al., 2011*; *Gupta et al., 2018*), leading to an asymmetrical shift in overall brain mass. Moreover, finding this in two separate populations suggests that these changes are common principles for fish evolving in a cave environment.

Early brain patterning is tightly regulated by genetic networks and developmental mechanisms that include organizing centers conserved across bilaterians (*Sylvester et al., 2011*; *Stoykova et al., 2000*; *Blaess et al., 2008*; *Denes et al., 2007*; *Holland et al., 2013*). These networks and mechanisms are controlled by both morphogen pathways, such as sonic hedgehog (shh) and bone morphogenic protein (bmp), and transcription factors (*Molina et al., 2007*; *Sasagawa et al., 2002*; *Wilson and Maden, 2005*), like the *homeobox gene* cluster (*hox*) (*Hunt et al., 1991*; *Krumlauf et al., 1993*; *Spitz et al., 2001*; *Hatta et al., 1991*), that orchestrate axis development and help regulate regional specification. Early forward genetic screens in model organisms led to the discovery of axial patterning genes, with mutants displaying drastic phenotypes impacting dorsal-ventral and anterior-posterior axis patterning. For instance, mutations in the transforming growth factor-beta and sonic hedgehog signaling pathways, in zebrafish, fly, and mice revealed severe phenotypic impacts in ventral forebrain development (*Chiang et al., 1996*; *Maity et al., 2005*; *Schier et al., 1996*). Our analyses suggest that two independently derived cavefish populations exhibit changes in early brain development that impact the majority of brain regions in a strictly dorsal-ventral fashion. While the divisions of the dorsal ventral axis are initially established via canonical pathways, such as *shh* and *bmp*, disruptions in downstream targets tend to be localized to specific ventral and dorsal regions (*Shimamura and Rubenstein, 1997*; *Ko et al., 2013*; *Karaca et al., 2015*; *Portella et al., 2010*; *Diaz and Puelles, 2020*). Therefore, our hybrid experiments may be the result of changes upstream of larger gene regulatory networks, impacting several genes that contribute en masse to the development of the dorsoventral axis (*Levine and Davidson, 2005*; *Alexandro et al., 2021*). This hypothesis would explain the concomitant dorsal contraction and ventral expansion revealed in the correlation and clustering analyses of surface to

cave hybrid larva. Further genetic and embryonic analyses will be needed to answer many outstanding questions, including: do these anatomical changes reflect a developmental 'hotspot', and are these anatomical features a common outcome for evolving non-visual sensory dominance?

Additionally, degeneration of the eye and subsequent impact on the entire brain has not been extensively studied in cavefish. While eye formation begins in cavefish populations, the embryonic eye primordia quickly undergoes apoptosis (*Jeffery and Martasian, 1998*; *Yamamoto and Jeffery, 2000*), followed by axon degeneration and denervation of retinal ganglion cells in the midbrain (*Soares et al., 2004*). This loss of afferent ocular contacts contributes to a decrease in the overall growth and size of the optic tectum (*Soares et al., 2004*; *Pottin et al., 2011*). Furthermore, rescuing eye development via lens transplantation from a surface donor to a cave host results in increased tectal mass on the contralateral side of the transplanted lens (*Soares et al., 2004*). Recent work has also hypothesized that changes in spatiotemporal gene expression of anterior neural markers impact both the eye and brain in a pleotropic manner (*Menuet et al., 2007*; *Pottin et al., 2011*; *Alié et al., 2018*). Therefore, a remaining question is whether eye degeneration and changes in dorsal-ventral patterning are separate mechanisms impacting the brain, or share a common mechanism that results in the overall ventral expansion and dorsal contraction observed in these cavefish populations. While this study did not resolve these questions, we are currently utilizing our novel computational atlas to determine the overall impact of eye degeneration on brain-wide anatomy.

Past comparative neuroanatomical research has raised many unanswered questions for how neural circuits evolve, including how variation in brain development between closely related groups relate to functional convergence of specialized brain regions (*Schumacher and Carlson, 2022*; *Emery and Clayton, 2004*; *Northcutt, 2002*; *Güntürkün, 2012*; *Earl, 2022*). For instance, it was initially thought that the pallial regions in lobe finned fishes (including tetrapod's) and ray finned fishes evolved independently to produce convergent functional traits, evinced by variation in developmental processes, cell types, and circuitry involved in these emergent behaviors. However, recent work utilizing a well-preserved fossil of an ancient ray finned species suggests that ray finned fish initially possessed the same developmental processes as lobbed finned fish (*Figueroa et al., 2023*). Therefore, functional similarity of the pallium may be a homologous ancestral state that was maintained by the two groups and not a convergent feature derived independently. Although this recent finding supports a major revision of our understanding of vertebrate brain evolution, we agree with the field that variation in cell type diversity and complexity of forebrain circuit development across these derived extant groups present unique and non-overlapping neurological traits. Neuroanatomical discoveries like this provide a prime example that creative strategies, including examination of the fossil record for soft-tissue preservation and functional studies in a diversity of non-model organisms, will be necessary to reveal unique and generalized principles of neural evolution.

Recent neural evolutionary studies in several non-model species have shown that some brain regions provide evolutionary potential for convergent function (*Schumacher and Carlson, 2022*; *Carlson, 2016*; *Earl, 2022*). For example, several independently derived groups of weakly electric fish have convergently evolved electrogenerative and electroreceptive potentials through expansions of the cerebellum (*Schumacher and Carlson, 2022*). This rather specific structural and functional innovation suggests that the cerebellum provides an anatomical substrate, with specific gene regulatory networks and cell types (*Güntürkün, 2012*), that are best suited for fish to gain electroreceptive properties (*Schumacher and Carlson, 2022*). While convergent functional innovations are observed in these independent lineages, the secondary consequences on behavior can vary from one species to another, suggesting that the process of evolution is acting upon functional expansion (electric properties), providing a substrate for novel behaviors (*Schumacher and Carlson, 2022*; *Carlson, 2016*). In our study, two independently derived cavefish populations show convergent neuroanatomical variation across the dorsal-ventral axis, resulting in an overall expansion of specific sensorimotor regions, including the hypothalamus and subpallium. Both the hypothalamus and the subpallium have diverse functions, and many behavioral modifications in cavefish, including aggression, stress, and sleep, have been related to functional variation in these areas (*Rodriguez-Morales et al., 2022*; *Chin et al., 2018*; *Jaggard et al., 2018*). We hypothesize that the reduction of dorsal regions preserves the energy needed to expand this ventral substrate of non-visual sensorimotor regions as anatomical potential to engender novel behaviors (*Moran et al., 2014*). That these changes are found in independently evolved populations further supports this notion.

Other taxonomic groups have also experienced increases in anterior forebrain volume, leading to the formation of new cell types and layers (*Woych et al., 2022*; *Lust et al., 2022*; *Briscoe et al., 2018*). The functional expansion of the forebrain and flexibility of supramodal cognition in primate brains has been linked to convergent adaptations to specific subsets of the cortex (*Sneve et al., 2019*; *Hill et al., 2010*; *Chaplin et al., 2013*). These changes included an expansion to the size and organizational complexity of the cortex, while also developing novel forebrain circuits that permit a functional compacity to produce more complex cognitive and social behaviors. Our data points to a similar phenomenon, wherein subpallial and hypothalamic regions are expanding in these two cavefish populations that likely shifts the primary integrative processes in the optic tectum to the ventral forebrain. It will be paramount going forward to determine whether ventral anatomical expansion is leading to new cell types, and how these anatomical changes impact ancestral neural circuits in relation to cavefish behavior.

In addition to volume, evolutionary changes in shape of neuroanatomical regions have been shown to alter function of different regions. The mammalian cortex, for example, has evolved from a smoother lissencephalic cortex in more ancestral species to a folded one in more derived animals such as primates (*Herculano-Houzel et al., 2014*; *Elias and Schwartz, 1969*; *Molnár et al., 2014*). Folding of the cortex is thought to increase surface area and has been implicated in more complex processing of the brain (*Molnár et al., 2014*; *Tallinen et al., 2014*; *Hofman and Falk, 2012*; *DeCasien et al., 2017*; *Abzhanov et al., 2006*). However, we do know that shape variation has been shown to be a common adaptation in other tissues. Beak differences in Galapagos finches have been shown to change in accordance with the size of food sources, and such changes have been shown to rely on differences in bone morphogenic protein signaling (*Parsons and Albertson, 2009*; *Kozol et al., 2021*; *Choi et al., 2016*). Craniofacial differences in African cichlids also have been shown to vary as an adaptive quality to food availability (*Gupta et al., 2018*; *Conith et al., 2019*). Furthermore, standard methods for assessing complex shape features have been applied to studying brain shape evolution in non-model organisms, generating anatomical evolutionary hypotheses that have lacked an appropriate model for assessing functional mechanisms of anatomical evolution (*Reardon et al., 2018*; *Gómez-Robles et al., 2014*; *Molnár et al., 2014*). By applying these morphological measuring and analyzing methods with our hybrid volume pipeline, we were able to see that complex shape phenotypes are likely genetically encoded, evidenced in hybrid intermediate phenotypes, and that similarities in covariation of shape and volume across dorsal and ventral regions may be impacted by shared mechanisms. However, due to the labor-intensive nature of shape analyses, we acknowledge that only 4 of 13 brain regions from the volumetric covariation analysis were assessed and are working to compare the remaining regions in our ongoing studies. Additionally, some of our shape variation could be capturing biological elements that are captured in the volumetric analysis, which we cannot rule out in the current analysis. While the functional and adaptive significance of differences in shape are not known, future work relating neuronal activity and function with differences in shape in this model could help address this question.

Together, these results support the developmental constraint hypothesis of brain evolution in cave-adapted *A. mexicanus* fish populations, suggesting early genetic and developmental impacts reshaping neuroanatomy brain-wide. This study represents the first computational brain atlas for a single species with multiple evolutionary derived forms, and the application of the atlas to hybrid animals represents the first assessment of how different neuroanatomical areas evolved in both volume and shape. Moreover, we can now combine this atlas with a myriad of cutting-edge tools that we have generated for this model, including functional neuroimaging and genome editing, that will allow researchers to identify the genetic mechanisms that explain these changes. The strong genetic and neuronal conservation of the vertebrate brain, as well as the simplified nervous system of fish, suggests that this model offers great potential to discover the general principles of evolution that impact the brain.

## Materials and methods
### Fish maintenance and husbandry
Mexican tetras (*A. mexicanus*) were housed in the Florida Atlantic Universities Mexican tetra core facilities. Larval fish were maintained at 23°C in system-water and exposed to a 14:10 hr light:dark cycle.

Mexican tetras were cared for in accordance with NIH guidelines and all experiments were approved by the Florida Atlantic University Institutional Care and Use Committee protocol #A1929. *A. mexicanus* surface fish lines used for this study; Pachón cavefish stocks were initially derived from Richard Borowsky (NYU); surface fish stocks were acquired from Rio Choy stocks. Surface Rio Choy were outcrossed to Pachón to generate $F_1$ hybrids, while $F_1$ hybrid offspring were incrossed to produce $F_2$ hybrids.

## IHC and imaging

Larval IHC was performed as previously published (*Avants et al., 2011*), using antibodies raised against total ERK (ERK; p44/42 MAPK [ERK1/2], #4696, Cell Signaling Inc, Danvers, MA, USA), Islet-1 and Islet-2 homeobox (Islet1/2, #39.4D5, Developmental Studies Hybridoma Bank, University of Iowa, Iowa City, IA, USA), TH1 (AB152, Sigma-Aldrich Inc, Burlington, MA, USA), and 5-HT (AB125, Sigma-Aldrich Inc). IHC-stained larvae were imaged on a Nikon A1R multiphoton microscope, using a water immersion 25×, NA 1.1 objective.

## Combined IHC and HCR in situ hybridization

To combine IHC and HCR in situ hybridization, the HCR in situ hybridization methodology for zebrafish embryos and larvae from Molecular Instruments (*Kozol et al., 2022*) was performed with the following exceptions: during the detection stage, larvae were incubated in probe solution for 48 hr to improve hybridization of RNA probes, and larvae were washed with 5× SSCTx (0.2% Triton X-100) instead of 5× SSCTw (0.2% Tween20) following hairpin incubation. Following HCR in situ hybridization, larvae were incubated in 5× SSCTx (0.2% Triton X-100) with 2% bovine serum albumin (BSA) at room temperature for 2 hr on a rocker (low speed). Following incubation, a primary antibody solution was added that included 5× SSCTx, 1% DMSO, 1% BSA, and 1:250 dilution of total-ERK antibody. Larvae were then incubated in primary antibody solution at 4°C on an orbital shaker set to 90 RPM for 48 hr. Primary antibody solution was then washed out three times with 5× SSCTw (0.2% Tween-20) for 10 min at room temperature on a rocker (low speed). Following primary incubation, a secondary antibody solution was added that included 5× SSCTw 1% DMSO, 1% BSA, and 1:500 dilution of goat anti-mouse IgGγ1 secondary antibody, Alexa Fluor 555 (Thermo Fisher, Waltham, MA, USA). Larvae were then incubated in secondary antibody solution at 4°C on an orbital shaker set to 16 hr and 90 RPM. Finally, secondary antibody solution was washed out three times with 5× SSCTw for 10 min at room temperature on a rocker (low speed) and subsequently imaged on a Nikon A1R confocal microscope, using a water immersion 20×, NA 0.95, long working distance objective, with 1.2× zoom.

## Generation of the brain-wide *A. mexicanus* atlas

To generate a segmented atlas for cave and surface *Astyanax*, we used a previously published neuroanatomical atlas (*Gupta et al., 2018*) from a related fish, the common zebrafish (*Danio rerio*), that is neuroanatomically homologous with *A. mexicanus* (*Jaggard et al., 2020*). We first modified the zebrafish brain browser brain atlas, neuropil, and cell body mask for the existing zebrafish resource CobraZ by using previously published Advanced Normalization Toolbox (ANTs; *Gupta et al., 2018*; *Wile, 2005*) registration and inverse registration scripts. This process creates a set of computational instructions for aligning our hybrid standard brain to the zebrafish reference brain, these instructions are then reversed to map the zbb segmented atlas onto our hybrid standard brain (*Figure 1b*). This created a hybrid brain atlas that could be used to register brains from all four *A. mexicanus* populations, producing a single computational atlas for measuring brain size and shape (*Figure 1c*, *Figure 1—figure supplement 1c–e*). We validated our *Astyanax* segmented atlas using three distinct approaches, cross-correlation of tERK saturated pixels, automated to hand segmentation overlap, and molecular markers that label distinct neuroanatomical regions. The cross-correlation analysis between registered *Astyanax* brain and the zbb reference brain revealed that the two were highly correlated (rho = 0.95). Next, we hand-segmented five brains each for surface fish, Pachón cavefish, and $F_2$ surface × cave hybrid larvae. These labeled neuroanatomical areas were then compared to automated segmentation from our brain atlas by running a custom cross-correlation script. 3D volumetric images were imported into MATLAB using the 'imread' function, vectorized to a 1D vector using 'imreshape', and then a Pearson's correlation was performed using the 'corr' function (scripts are deposited in the Dryad depository for this study; *Schlager, 2017*). The automated segmentation to

**Table 1.** Hybridization chain reaction (HCR) in situ hybridization probes.

| Gene | *A. mexicanus* population | Ensembl ID | Molecular Instruments Lot # |
|------|---------------------------|------------|------------------------------|
| *gbx1* | Surface | ENSAMXT00000037099.1 | PRO705 |
| *gbx1* | Pachón | ENSAMXT00005023309.1 | PRO706 |
| *otx2b* | Surface | ENSAMXT00000055482.1 | PRQ451 |
| *otx2b* | Pachón | ENSAMXT00005060650.1 | PRQ452 |
| *oxt* | Surface | ENSAMXT00000041101.1 | PRQ449 |
| *oxt* | Pachón | ENSAMXT00005006990.1 | PRQ450 |

hand segmentation analysis revealed no difference in segmentation accuracy across *Astyanax* populations and >80% correlation between ERK-defined hand-segmented and automated segmentation (*Figure 1—figure supplement 2*). Finally, antibodies and RNA probes were used to test the accuracy of segment bounding for subregions that are known to outline specific molecularly defined neuronal populations (*Figure 1d*, *Figure 1—figure supplement 3*, *Table 1*).

## Automated segmentation and brain region measurements

Surface, Pachón, and surface to Pachón hybrid larval tERK-stained brains were registered and segmented using the aforementioned ANTs scripts. The resulting brain mask and segmentation file for each larvae was then processed using the morphometric analysis suite CobraZ. CobraZ measures the size of segmented regions of the brain and calculates regional size as percent of total brain (pixels of brain region/total pixels in brain; *Gupta et al., 2018*). We did not amend ANT's registration or inverse registration scripts, nor did we change the CobraZ parameters used in *Gupta et al., 2018*. We did additionally produce a modified segmentation file that defines larger subregions that overlap with tERK neuropil to provide cross-correlation analysis across brain regions and populations. Finally, we tested the accuracy of subregion segmentation using the HCR in situ hybridization probes and antibodies previously mentioned in the above sections. All scripts used in the analysis and generation of statistics (Supplementary Statistical Tables.xlsx) and materials in figures are archived in the Dryad submission associated with this study (see Data sharing; *Schlager, 2018*).

## 3D geometric morphometric methods to characterize shape variation

Correlations between volumes of brain regions were determined using custom-written scripts in Python. Volume data was imported from Microsoft Excel into Python using the pandas library. SciPy was then used to determine the pairwise correlation between all brain regions. The Seaborn library was then used to generate a heat map with annotations set to 'True' to overlay correlation coefficients on the pairwise correlation matrix. Cluster analysis of the corresponding pairwise correlation matrix was performed using SciPy toolkit. The distance matrix was first calculated from the correlation matrix and then indexed into the corresponding clusters. The correlation matrix was then clustered by grouping all regions that clustered (i.e. had the same index value). The resulting metric was again generated using Seaborn. The code for these analyses can found in the Dryad repository (see README, *Kozol et al., 2021*).

## 3D geometric morphometric methods to characterize shape variation

We used 3D geometric morphometrics to characterize shape variation in six different brain regions: preoptic, pineal, cerebellum, hypothalamus, tectum, and tegmentum. For the preoptic and pineal regions we landmarked parental and $F_2$ hybrid populations (preoptic: Pachón [n=23], surface [n=23], $F_2$ [n=34]; pineal: Pachón [n=15], surface [n=23], and $F_2$ [n=36]). We landmarked only $F_2$ hybrids for the remaining brain regions (cerebellum, n=28; hypothalamus, n=34; tectum, n=30; tegmentum, n=34). We used a combination of landmark types to best assess shape in each brain region (i.e. fixed, semi, surface), and placed landmarks onto the extracted 3D meshes using the morphometrics program LandmarkEditor (v3.0) (*Baken et al., 2021*). Landmark placement was manually conducted, except for the pineal, in which we utilized a semi-automated method (see below). To characterize shape in

the remaining brain regions we used 16 landmarks for the preoptic (fixed LM n=16), 102 landmarks for the cerebellum (fixed LM n=2; surface semi-landmarks n=99), 34 landmarks for the hypothalamus (fixed LM n=34), 202 landmarks for the tectum (fixed LM n=2, surface semi-landmarks n=200), and 26 landmarks for the tegmentum (fixed LM n=26).

For the pineal body, we placed two fixed landmarks at the anterior and dorsal apexes of the pineal and surrounded the base of the pineal with 26 sliding semi-landmarks. We then took advantage of a procedure to automate the placement of 99 surface landmarks across the pineal region to wrap the pineal body with sliding surface semi-landmarks to best characterize the shape of this subregion among individuals. This required building a computer-aided design (CAD) template of the pineal using FreeCAD (v.0.16.6712), which we modeled as a hemisphere, and placing the fixed landmarks, sliding semi-landmarks (*Figure 5—figure supplements 1 and 2*), and surface landmarks on the CAD model using LandmarkEditor. We then used the R package Morpho to map the surface landmarks from the template to the pineal model of each individual specimen using the placePatch function (*Adams, 2021*; *Conith et al., 2023*).

Following landmark placement, we performed a Procrustes superimposition on our shape data for each brain region to remove the effects of translation, rotation, and scaling from all individuals using the gpagen function from the geomorph (v4.0) package in R (*Schlager, 2017*; *Schlager, 2018*). Following superimposition, we performed a PCA to reduce our landmark data to a series of axis that best reflect differences in brain shape variation with each region. We plotted the component scores from each PCA to visualize how the shape of each brain region varies among parental populations and/or within the $F_2$ hybrids (*Figure 5—figure supplements 3–6*). We also extracted PC1 – the PC that explains the greatest amount of shape variation for a given brain region – from the cerebellum, hypothalamus, tectum, and tegmentum for use in a subsequent cluster analysis.

## Allometry

We explicitly wanted to retain the allometric component of shape variation given that one of our major goals was to understand how a variable related to size – volume – varies among brain regions and populations. Similarly, developmental modularity may be a function of allometric scaling relationships (*Conith et al., 2023*) so by retaining allometry in our shape data, the results from our volumetric and shape cluster analysis should be more comparable. Despite this, we tested for allometry in our shape data by performing a multivariate regression of shape on centroid size using the procD.lm r function from geomorph and found three of our $F_2$ surface × cave hybrid brain regions exhibited an association between shape and size (cerebellum, hypothalamus, tegmentum), while three did not (pineal, preoptic, tectum), further highlighting the complex nature of size and shape relationships within the brain (*Figure 5—figure supplement 6*).

## Partial least squares and cluster analysis using shape data

To assess the degree of association between brain subregion shape and volume, we performed a multivariate regression of shape on volume using the procD.lm r function from geomorph. Similarly, to assess associations among brain subregion shapes, we performed a PLS analysis using the two.b. pls r function from geomorph. PC1 extracted data from the *3D geometric morphometric methods to characterize shape variation* section were then run through the pairwise correlation and cluster SciPy functions described for the $F_2$ hybrid volumetric analyses.

## Statistics

All wildtype population standard t-tests were calculated using the program CobraZ (*Bradic et al., 2012*). For hybrid population comparisons, Prism (GraphPad Software Inc, San Diego, CA, USA) was used to run standard ANOVAs, followed by a Holm-Šídák's multiple comparisons test to correct for comparing across statistical permutations for each figures analysis. All statistical tables for main figures and figure supplements are available in the 'Supplementary Statistical Tables'. To evaluate covariation of $F_2$ subregions, geometric morphometry analyses were all conducted in R (*Choi et al., 2016*; *Avants et al., 2011*) using the packages geomorph (v4.0) and Morpho (v2.6) (*Kozol et al., 2022*; *Wile, 2005*; *Schlager, 2017*; *Schlager, 2018*) to assess associations and produce morphospace plots. Sample sizes for this study were based off previous studies (*Bradic et al., 2012*; *Sittaramane et al., 2009*), and therefore power analyses were not conducted.

## Data sharing

All data, statistical tables, custom code, and adapted tools have been made available on a Dryad repository, https://doi.org/10.5061/dryad.w9ghx3frw/ (*Choi et al., 2016*).

## Acknowledgements

We would like to thank Dr. Harry Burgess for his help in adapting the zebrafish brain browser atlas and modifying files for CobraZ to analyze *A. mexicanus* neuroanatomy. We thank Dr. James Jaggard for his expertise and early atlas work as a foundation for this project. To the administration and staff at the Jupiter Life Science Initiative in the Department of Biology at Florida Atlantic University, especially Peter Lewis and Arthur Loppatto for overseeing the health and care of the FAU Astyanax fish facility. This research was supported by grants from the NIH to ERD R15MH118625-0, ACK HFSP-RGP0062/1R01GM127872/, JEK R35GM138345/R15HD099022, ACK and JEK R21NS122166. This work was also supported by a grant from the NSF to ERD, JEK, and ACK #1923372 and an NSF grant to JEK 2202359.

## Additional information

### Funding

| Funder | Grant reference number | Author |
|---|---|---|
| National Institutes of Health | R15MH118625 | Erik R Duboue |
| National Institutes of Health | R01GM127872 | Alex C Keene |
| National Institutes of Health | R35GM138345 | Johanna E Kowalko |
| National Institutes of Health | R15HD099022 | Johanna E Kowalko |
| National Institutes of Health | R21NS122166 | Johanna E Kowalko Alex C Keene |
| National Science Foundation | 1923372 | Johanna E Kowalko Alex C Keene Erik R Duboue |
| National Science Foundation | 2202359 | Johanna E Kowalko |
| Human Frontier Science Program | RGP0062 | Alex C Keene |
| National Institutes of Health | DE026446 | Craig Albertson |

The funders had no role in study design, data collection and interpretation, or the decision to submit the work for publication.

### Author contributions

Robert A Kozol, Conceptualization, Data curation, Formal analysis, Investigation, Methodology, Writing - original draft, Project administration; Andrew J Conith, Data curation, Formal analysis; Anders Yuiska, Formal analysis; Alexia Cree-Newman, Bernadeth Tolentino, Kasey Benesh, Alexandra Paz, Evan Lloyd, Data curation; Johanna E Kowalko, Alex C Keene, Conceptualization, Funding acquisition, Writing - original draft, Project administration; Craig Albertson, Conceptualization, Writing - review and editing; Erik R Duboue, Conceptualization, Investigation, Writing - original draft, Project administration

### Author ORCIDs

Erik R Duboue  http://orcid.org/0000-0003-3303-5149

### Ethics

Mexican tetras were cared for in accordance with NIH guidelines and all experiments were approved by the Florida Atlantic University Institutional Care and Use Committee protocol #A1929.

### Decision letter and Author response

Decision letter https://doi.org/10.7554/eLife.80777.sa1
Author response https://doi.org/10.7554/eLife.80777.sa2

---

## Additional files

### Supplementary files

• MDAR checklist

• Source code 1. README.md file that includes text describing the material in *Source code 2*, a repository of scripts used in this study.

• Source code 2. Kozol et al. Code.zip file containing bash (.sh), fiji (.ijm), and matlab (.m) files used for this study.

### Data availability

All raw and analyzed data, custom code and adapted tools have been uploaded into a Dryad repository, https://doi.org/10.5061/dryad.w9ghx3frw. Custom code and adaptive tools are also included in the supplemental material.

The following dataset was generated:

| Author(s) | Year | Dataset title | Dataset URL | Database and Identifier |
|---|---|---|---|---|
| Kozol RA, Cree-Newman A, Tolentino B, Paz A, Yuiska A, Banesh K, Conith A, Lloyd E, Kowalko J, Keene A, Albertson C, Duboue ER | 2023 | Data from: A brain-wide analysis maps structural evolution to distinct anatomical modules | https://dx.doi.org/10.5061/dryad.w9ghx3frw | Dryad Digital Repository, 10.5061/dryad.w9ghx3frw |

---

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
