## [Editor Report]

The authors ask if brain regions change based on the functional constraints or developmental constraints. To address this, the authors introduce an automated method for brain segmentation based on the zebrafish tool to study brain evolution in Astyanax.

---

## [Decision Letter]

**Decision letter after peer review:**

[Editors’ note: the authors submitted for reconsideration following the decision after peer review. What follows is the decision letter after the first round of review.]

Thank you for submitting the paper "Automated anatomical mapping finds brainwide evolution occurring in distinct developmental modules" for consideration by *eLife*. Your article has been reviewed by 3 peer reviewers and the evaluation has been overseen by a Senior Editor. The following reviewers have agreed to share their identity (Reviewer 2: Masato Yoshizawa; Reviewer 3: Daphne Soares).

We are sorry to say that, after consultation with the reviewers, we have decided that this work will not be considered further for publication by *eLife*.

Comments to the Authors:

The authors tackle the role of functional and developmental constraints in brain evolution, an important and long-standing question. The approach nicely tests the developmental constraint hypothesis, which was highlighted in their correlation studies of volumetric and shape change, but is less successful with functional constraints. The reviewers appreciate the advance of automated brain characterization including shape changes with the large quantity data on brain shape-genetics but raised questions about the ability to reliably detect smaller subregions. They noted the need to add additional data on dark-raised individuals and also raised questions about novelty. Given these issues and the amount of additional work it requires, the paper seems more appropriate for a specialized journal. Please see the detailed review comments below.

*Reviewer #1 (Recommendations for the authors):*

To meet criteria for *eLife*, I would expect that the issues detailed should be addressed, and the following.

Additional external validations of the atlas should be provided. Molecular markers would be ideal for this process.

To identify whether the adaptation to cave environments proceeds by generalizable paths, an atlas for an unrelated Astyanax lineage should be compared.

L153 and Figure 4d,g. What is the statistical support for labeling Cluster 3 as an independent group? Is the node that branches cluster 3 from 2 statistically significantly supported?

Figure 4c. Does mapping the 6 clusters onto a brain reveal regional similarities in co-evolving region sizes?

Figure S5 legend is too sparsely worded to be clear. What the red circles represent should be clearly stated. What do the A and B and X and Y variables represent? These are not defined. "PC1 describes preoptic width, while PC2 describes length." Correct me if I'm wrong, but here I think you mean something more subtle than what's stated. Shouldn't this be something about the loading of these PCs having an over representation of width/length distances. If the simple statement is true, why not report the absolute width and length in addition to the PCA?

Figure 5. As commented above, there is a conflation of PC with a simple length measurement. The optic tectum varies on PC1, and you show it is longer. How much of PC1 is explained by length, for example. Why not provide the length measurement?

Line 198-200. The conclusions from this paragraph are too strong. One issue is why one must invoke developmental constraints in the patterning of the brain. It may be true that expansion of one brain region comes at the expense of another, but these data do not demonstrate this.

Figure 5c,d. The differences between these two graphs are trivial, as I understand them, and there seems to be no new information provided by using two graphs. Why not just provide 5d, with regions labeled on the Y-axis?

Line 219-220. This conclusion is not supported by a direct analysis. Volume covariation and shape covariation are not directly compared. Rather, in the volumetric analysis, only 4 regions are analyzed, rather than 13 for the shape analysis. These should be analysed in the same way in order to make this statement. But even this would be simply a correlation.

But further, could a similar covariation (if found) be simply a result of the fact that volume is a function of the parameters (e.g., length, width) analysed during shape analysis? As a result, shape and volume are not orthogonal variables for which a finding of distinct developmental mechanisms would be compelling.

Line 260 and elsewhere. "Our data show that the dorsal-caudal areas of the brain evolve together…" This work shows that in the Pachon line has evolved in this way. The sample size is derived from F2 animals from a outcross of this line, and as such the separate animals represent unique genetic combinations of Pachon alleles, but it is from a single evolutionary trajectory. As such, additional examples of evolution (for example from another cave) may operate by changing other brain region sizes. I see that this caveat is then raised (L 266), but the conclusions remain overstated in this paper elsewhere.

*Reviewer #2 (Recommendations for the authors):*

This is not the authors' fault, but there is confusion between these two hypotheses in general – both (functional and developmental) can be regulated by genes. Developmental constraints can act brain-wide and local (like islet_1/2_ expressing domain). Functional constraints can also act on local-nearby regions but also long distances such as between the cerebrum and cerebellum (co-evolution of cerebrum and cerebellum were presented as an example of the functional constraints). Thus, brain-wide or not is not such a good criterion to argue these two hypotheses although Montgomery et al., 2016 mentioned. To me, these two hypotheses are proxies of the question of whether the brain regions are regulated by developmental genes (Pax6, Hoxs, Otx etc) vs. functional axonal/synaptic genes (Netrin, semaphorin, GluN receptors etc). The authors briefly mentioned that cluster3 (subpallium and dorsal diencephalon) could be an example of the functional constraints but it was with not so positive reasons; that is, this region was not constrained by other regions (L258-259). If the authors would like to mention the functional constraints, I feel they need to show positive data, such as neural activities are not tightly associated with other brain regions compared with between the clusters 1 and 2.

My suggestion is that the authors may need to restate their research question to align what they have in the data.

here are the other points I am concerned about:

L34-39: "Two central hypotheses are thought to drive anatomical brain evolution;…"

I am afraid that the sentence for the developmental constraint hypothesis is a little misleading by stating "..change together in a concerted matter." many studies showed the brain develops in a mosaic fashion. I guess the authors would like to state as "the developmental constraints hypothesis suggests that most of the individual brain components tend to evolve together" as Montgomery et al., 2016 stated.

Also the authors mentioned 'mosaic' relating to the functional constraints hypothesis, yet, in developmental biology, 'mosaic' is also used for developmental processes. I suggest the authors clarify whenever they use 'mosaic' for functional or developmental.

L101: "…brain regions (Figure 2 – Supplement 1a and b)"

need reference

Figure 4e and 4f, the figures were swapped according to the Main body text and Figure 5 figure legend.

Figure 5—figure supplement 1, difficult to understand where the landmarks were from. Please superimpose the preoptic area's image on the landmarks.

I do not see if the original image data and other related data will be shared or not.

*Reviewer #3 (Recommendations for the authors):*

I thoroughly enjoyed this manuscript, I think it can eventually be a fantastic paper! I would be very happy to look at it again and I have no problems with the data. Please understand that my criticism is basically saying that I don't think you completely know HOW interesting/impactful your results are because you do not address to the long arguments and disagreements in the literature. I think you should spend a while really dissecting what people have been thinking about the evolution of vertebrate brain. Read the vast body of literature on brain evolution, the works of William Hodos, Ann Buttler, Georg Strieder and Glenn Northcutt especially come to mind right away. Your results based on their original work can be that much more insightful. These established authors have raised many questions which you can chime in with your new techniques. It will be a wasted opportunity to just argue that astyanax is a good model, show the reader WHY it is a good model, what do the results mean. I think then it will elevate the paper to an *eLife* level.

I am including one specific set of points on the abstract that illustrate the same issues again and again in the entire text (I'll be blunt because it is easier, keep in mind that I am a fan of the work):

"Brain anatomy is highly variable and it is widely accepted that anatomical variation impacts brain function and ultimately behavior."

I think the initial statement of the abstract is already confounding. You are already not bringing evolutionary neuroscientists to your side. It is too simplistic. I think anyone who knows anything about the issue will either dismiss you, thinking that you have not done your homework in terms the literature, or that your knowledge of the subject is not sophisticated. There is an enormous body of evidence on the literature that show how conserved brains actually are. Diversity and homology are actually one of the main tenants of MANY brain evolutionary studies. Also revise the sentence to avoid repetition of "anatomy/anatomical" for stylistic reasons.

"The structural complexity of the brain, including differences in volume and shape, presents an enormous barrier to define how variability underlies differences in function."

I respectfully disagree, the challenge is no longer in this level of analysis. Neurons themselves, circuits and modulation are the next frontier that will inform our questions. I think the first 2 sentences are weak and do not buttress the argument of why this study is important and novel. I would recommend to rewrite it.

"In this study, we sought to investigate the evolution of brain anatomy in relation to brain region volume and shape across the brain of a single species with variable genetic and anatomical morphs."

Again this has been done for years, at this point of reading, I am not convinced why this is novel enough to give insights to the audience of *eLife*. I think here you really have the chance to drive your point that your results are really important, but you don't yet have the knowledgeable reader on your side.

[Editors’ note: further revisions were suggested prior to acceptance, as described below.]

Thank you for resubmitting your work entitled "Automated anatomical mapping finds brainwide evolution occurring in distinct developmental modules" for further consideration by *eLife*. Your revised article has been evaluated by Marianne Bronner (Senior Editor) and a Reviewing Editor.

The manuscript has been improved but there are some remaining issues that need to be addressed, as outlined below:

1. The authors must check whether the eye-size is a confounding factor of the brain region volume changes. This could be done by a correlation analysis of eye-size vs brain regions/shape (PCA1 or PCA2 axis value).

2. The discussion and references on brain evolution need to be significantly expanded.

3. Please attend to the more minor revisions raised by the reviewers.

*Reviewer #1 (Recommendations for the authors):*

Thank you to the authors for their thoughtful responses to my concerns. I find the manuscript to be significantly improved. I feel that the arguments they make are now well-supported. I have a few relatively minor comments on places where the writing was not clear.

Is the segmentation by tERK with labels for region-specific markers done with ONLY the tERK information? Is there any chance that signal from the regional marker could influence the segmentation?

Lines 126-132. The figure references do not refer to the correct panels (wrong genes). Also, anti-TH is not present in any figure; please add this.

*Reviewer #2 (Recommendations for the authors):*

This is a significant and excellent update to the original submission by Kozol et al. Most of the reviewers' concerns are resolved with a much improved clearer logical flow and newly added Molino cavefish and F2 hybrid data, further adding detailed shape and volume analyses. There are one major concern and several minor points in this resubmitted version of their manuscript.

Although the volume change analyses were now more elaborated, it is unclear if the eye size is a correlated factor to explain the antagonistic volume-changes between the dorso-caudal and ventro-rostral brain regions. The authors explained that this pattern of volume change is the result of developmental/genetic constrain with new F2 data, which is convincing. Yet, the major sensory difference at 6 days old – functional vs non-functional eyes (surface fish and cavefish, respectively) – could affect the brain anatomical changes. I assume that authors took whole body pictures before dissection (or, eyes were scanned together during brain imaging) and the eye-size parameter is available to test whether the whole brain landscape is basically correlated with the eye size. None of the other sensors including the olfactory epithelium, lateral line, and inner ear, was reported with significant differences between surface fish and cavefish, as far as I know. I believe the eye size is the important factor in arguing the brain architecture while I was reading through this manuscript, I assumed the brain developmental genes (otx, islet, emx) might regulate these changes but eye developmental genes (pax 6, lens genes) could be the major driver of these brain morphological changes, instead. To answer their research question, whether the whole brain regions or individual brain regions evolved, I believe the eye is the important brain region the authors should consider. F2 hybrids would be a great resource to check it.

---

## [Author Response]

[Editors’ note: The authors appealed the original decision. What follows is the authors’ response to the first round of review.]

Reviewer #1 (Recommendations for the authors):To meet criteria for eLife, I would expect that the issues detailed should be addressed, and the following.Additional external validations of the atlas should be provided. Molecular markers would be ideal for this process.

We agree that an addition of additional markers would make the manuscript stronger for publication. We have now added 5 additional markers. This involved not only adding additional antibody labels, but also devising an HCR protocol that was compatible with brain registration. This classically has been challenging as the harsh molecular washes for in situ hybridization often warp and distort brain tissue such that it cannot be correctly registered. HCR alleviated those concerns, and made for a high throughput strategy for labeling different molecular targets with known expression patterns. The addition of these markers show that the tested markers were bound to the segmented regions which were automatically computed by the program ANTs. Methods and results can be found in Figure 1, Figure 1 —figure supplement 1, and lines 105-118 and 422-440. We believe the addition of these experiments strengthens the validity of our atlas and will be incorporated into the downloadable atlas package.

To identify whether the adaptation to cave environments proceeds by generalizable paths, an atlas for an unrelated Astyanax lineage should be compared.

We agree that assessing another population of fish is a valuable resource, and is needed for claims about ‘generalizability.’ To address this concern, we have repeated our methods that were previously presented for surface fish and Pachón cavefish on another, independently evolved cavefish population, Molino. This is a unique cave, in that it evolved completely independent of the Pachón population in a second migration sweep of surface fish invading caves. Our findings in Molino with brain anatomy in pure populations as well as F_2_ hybrids parallels our findings in Pachón, suggesting that a ventral expansion and a dorsal contraction of the brain is found in multiple cavefish. These findings and discussions can be found in Figure 2, Figure 3, Figure 4 and lines 132-148, 151-165, 169-188, 189-210, and 289-294.

While claims about evolutionary generalization cannot be made without. an exhaustive assessment of various animals that span the kingdom, similar findings in independently evolved Astyanax morphs suggests that these patterns of brain evolution are likely a generalized feature in these Mexican cavefish.

L153 and Figure 4d,g. What is the statistical support for labeling Cluster 3 as an independent group? Is the node that branches cluster 3 from 2 statistically significantly supported?

The clustering in these studies were performed using hierarchical clustering, and while this approach is a widely used approach for clustering, it does not set the number of clusters, per se. The number of clusters from a hierarchical clustering approach can be as few as one (everything in one cluster) or as many as the number of individuals represented. For instance, the cluster maximum for the surface to Pachón hybrids was 6, while the surface to Molino hybrids was 12. Our cutoff was based on that branch point that gave the most distinct grouped clusters. In order to established statistical basis for calling these groups separate, we tabulated those R-values in each cluster and examined whether there was statistical significance suggesting that they were indeed different. Unsurprisingly, the t [13.82] and p-value [>0.0001] for cluster 2 and 3 revealed that the two were significantly different, suggesting the two groups are indeed different. We have added statistical tables for all comparisons to the supplemental material, Figure 4 —figure supplement 5-10.

Figure 4c. Does mapping the 6 clusters onto a brain reveal regional similarities in co-evolving region sizes?

We thank the reviewer for the question and have gone back to map our 6 clusters onto the Astyanax atlas. We originally only mapped back the large subdivisions of our atlas due to labor constraints, but have now written additional code that maps back brain regions for all 180 areas into their respective clusters. This not only allowed us to address this reviewers concern, but we hope that it serves as a resource for the community.

After mapping all 180 regions that fell into 6 clusters back onto the brain, we found the same trend as we did with our 13 brain region clusters, specifically that all 6 clusters are arranged in a dorsal ventral fashion and fell into groups of closely adjacent regions. We also found that dorsal to ventral comparisons of the 6 clusters also showed strong negative associations similar to those found across cluster 1 and 2 of the 13 brain region atlas. We then repeated this analysis in surface to Molino F_2_ hybrids and found that brain regions are grouped in 12 distinct dorsal to ventral clusters, with similar positive and negative relationships found in surface to Pachon F2 hybrids. We have now replaced Figure 4 with a new figure depicting the un-clustered and clustered arrays of the 180 brain regions, along with projection maps and coronal optical sections providing examples of the clustered regions. The previous Figure 4 has now been added to the supplemental data as Figure 4 – Supplement 1. Results and discussion text can be found in lines 169-188, 189-210, and 284-294.

Figure S5 legend is too sparsely worded to be clear. What the red circles represent should be clearly stated. What do the A and B and X and Y variables represent? These are not defined. "PC1 describes preoptic width, while PC2 describes length." Correct me if I'm wrong, but here I think you mean something more subtle than what's stated. Shouldn't this be something about the loading of these PCs having an over representation of width/length distances. If the simple statement is true, why not report the absolute width and length in addition to the PCA?

We apologize for the sparse wording and mislabeling of the figure axes. We have added additional descriptive information into the figure legend to assist a reader in understanding our PCA plots. We have also expanded this legend to more clearly state the data presented. We apologize for this oversight, and thank the reviewer for calling our attention to it.

Figure 5. As commented above, there is a conflation of PC with a simple length measurement. The optic tectum varies on PC1, and you show it is longer. How much of PC1 is explained by length, for example. Why not provide the length measurement?

The reviewer is correct that our presentation of these data was misleading. We meant to use PCA to determine statistical relationships between the 3d volumetric shape, and to show, broadly, how these varied in both x and y axis dimensions. However, our description in the text and generalized arrows along physical axes of the brain regions was a mistake. Our intent was not to conflate, but rather to make it easy for a reader to interpret. We have since edited Panels a and b in the figure, and added supplementary figures, Figure 5 —figure supplement 3-6, that provide wire diagrams and landmarks over the projected brain regions to elucidate the 3d complexity being analyzed.

In regards to the PCA, the PCA was generated by placing reference points at pre-defined areas on each neuroanatomical locus, applying 3-dimentional morphometrics, and then using principal component analysis of the high dimensionality data set to determine which principle components explained the largest amount of the difference between the two. The PCA showed the statistical relationship between morphs, and revealed that, as would be expected for a genetically determined trait, that F_2_ hybrids showed intermediary values that spanned the range of surface cave. Often times, we found that a given principal component aligned with an expansion/contraction along either the x- or y-axis. Other times, the PCA explained a curvy-linear axis, as was the case for the cerebellum (in Figure 5b), and we highlighted that. However, the reviewer is correct: because the PCA was performed on 3d morphometrics and not linear data sets, true relationships between 3d data and x- and y-axis cannot be made.

But further, could a similar covariation (if found) be simply a result of the fact that volume is a function of the parameters (e.g., length, width) analysed during shape analysis? As a result, shape and volume are not orthogonal variables for which a finding of distinct developmental mechanisms would be compelling.

3D shape analysis is certainly a different parameter than volume, yet the two are both continuous variables. We show that the two continuous values correlate beyond statistical significance and believe this suggests that the two are likely related to each other. We do believe, however, as I think the reviewer is stating, that we could be capturing some measure of volume within our shape parameter, and we now make this point in the discussion (lines 386-387).

Line 198-200. The conclusions from this paragraph are too strong. One issue is why one must invoke developmental constraints in the patterning of the brain. It may be true that expansion of one brain region comes at the expense of another, but these data do not demonstrate this.

We agree that the statement was too strong, in lieu of cluster mapping to demonstrate a concomitant tradeoff between dorsal-ventral poles of the brain. However, with the addition of the Molino cave analysis and subsequent mapping of hybrid brain clusters, we believe the data and analysis now supports this conclusion. We have edited the language in the text to now state “an anatomical tradeoff” that appears convergent across cave populations. These changes are now found in lines 167 and 187.

Figure 5c,d. The differences between these two graphs are trivial, as I understand them, and there seems to be no new information provided by using two graphs. Why not just provide 5d, with regions labeled on the Y-axis?

While these two graphs are very similar and we are happy to drop panel 5c we believe that for the sake of transparency, showing clustered and unclustered graphs are useful for the reader, and does not take away from the study. Again, we are happy to remove if needed, but have left unchanged as of now for transparency sake.

Line 219-220. This conclusion is not supported by a direct analysis. Volume covariation and shape covariation are not directly compared. Rather, in the volumetric analysis, only 4 regions are analyzed, rather than 13 for the shape analysis. These should be analysed in the same way in order to make this statement. But even this would be simply a correlation.

We agree with the reviewer that a direct comparison was not performed and have since softened the language throughout the paragraph (line 369-372). This includes pointing out that this is a first effort and will need to analyze the entire atlas in future work. Shape is significantly more laborious to assess, and we were not able to look at shape for every region. Moreover, we believe that the volume is a resource to the community, whereas shape was assessed to try to determine whether there was a relationship. We have also highlighted this discrepancy in the discussion (lines 418 to 420). We did however directly compare dimension reduced 3D morphometric values to volume in 6 regions of the 13 region atlas (Figure 5 —figure supplement 9 and lines 254-264). We hope these changes and the transparency of analytical terms satisfies the valid concerns of the reviewer.

Line 260 and elsewhere. "Our data show that the dorsal-caudal areas of the brain evolve together…" This work shows that in the Pachon line has evolved in this way. The sample size is derived from F2 animals from a outcross of this line, and as such the separate animals represent unique genetic combinations of Pachon alleles, but it is from a single evolutionary trajectory. As such, additional examples of evolution (for example from another cave) may operate by changing other brain region sizes. I see that this caveat is then raised (L 266), but the conclusions remain overstated in this paper elsewhere.

We agree that these statements were overstated without analyzing additional cave populations. We have since analyzed both pure Molino cavefish and surface to Molino cave F_2_ hybrids. Molino pure and hybrid data was nearly identical to that of Pachón cavefish and suggests this dorsal-ventral shift in brain mass is likely a generalizable trait for Astyanax mexicanus.

Reviewer #2 (Recommendations for the authors):This is not the authors' fault, but there is confusion between these two hypotheses in general – both (functional and developmental) can be regulated by genes. Developmental constraints can act brain-wide and local (like islet_1/2_ expressing domain). Functional constraints can also act on local-nearby regions but also long distances such as between the cerebrum and cerebellum (co-evolution of cerebrum and cerebellum were presented as an example of the functional constraints). Thus, brain-wide or not is not such a good criterion to argue these two hypotheses although Montgomery et al., 2016 mentioned. To me, these two hypotheses are proxies of the question of whether the brain regions are regulated by developmental genes (Pax6, Hoxs, Otx etc) vs. functional axonal/synaptic genes (Netrin, semaphorin, GluN receptors etc). The authors briefly mentioned that cluster3 (subpallium and dorsal diencephalon) could be an example of the functional constraints but it was with not so positive reasons; that is, this region was not constrained by other regions (L258-259). If the authors would like to mention the functional constraints, I feel they need to show positive data, such as neural activities are not tightly associated with other brain regions compared with between the clusters 1 and 2.My suggestion is that the authors may need to restate their research question to align what they have in the data.

These hypotheses are not singularly defined in the field, and interpretations of what developmental and functional means often varies from one usage to another. We agree with the reviewer that an argument for a “functional constraint” would be strengthened by functional experiments, and that arguments for developmental would be augmented by developmental genes such as Pax6 or Nkx2.2, while arguments for function would be bolstered by corresponding mechanisms involving ‘functional’ genes such as Netrin and semaphorins. While this is a long-term goal of the project, we are far away from having these results.

Due to these limitations, we have decided to remove the section arguing for functional constraint of cluster 3, and have tried to be more cautious in our arguments for these two hypotheses throughout the manuscript.

here are the other points I am concerned about:L34-39: "Two central hypotheses are thought to drive anatomical brain evolution;…"I am afraid that the sentence for the developmental constraint hypothesis is a little misleading by stating "..change together in a concerted matter." many studies showed the brain develops in a mosaic fashion. I guess the authors would like to state as "the developmental constraints hypothesis suggests that most of the individual brain components tend to evolve together" as Montgomery et al., 2016 stated.

The reviewer’s interpretation of our intent is correct, and we have tried to clarify the manuscript to better reflect that. The text now reads, “, that most of the brains regions tend to evolve together” (lines 37-42).

Also the authors mentioned 'mosaic' relating to the functional constraints hypothesis, yet, in developmental biology, 'mosaic' is also used for developmental processes. I suggest the authors clarify whenever they use 'mosaic' for functional or developmental.

We understand the reviewer’s confusion and have edited the text that describes the two hypotheses. We apologize for the vagueness and hope our revised text satisfies this reviewers concern. Additionally, the term mosaic was removed from the manuscript, resulting in the singular usage of functional constraint to describe this hypothesis.

L101: "…brain regions (Figure 2 – Supplement 1aandb)"need reference

The main body text has been updated to include the correct references. The text now reads, “(Figure 2 – Supplement 1aandb, Figure 2 —figure supplement 2-17) [49,50]." (line 347-348).

Figure 4e and 4f, the figures were swapped according to the Main body text and Figure 5 figure legend.

We have edited the main body text and figure legends to reflect changes in the figures themselves. Figure 4 is now a cluster analysis of the two hybrid populations. The initial Figure 4 is now Figure 4 – Supplement 1, with the addition of the Molino cavefish hybrid population data.

Figure 5—figure supplement 1, difficult to understand where the landmarks were from. Please superimpose the preoptic area's image on the landmarks.

We apologize for leaving out the preoptic area’s image. We have edited the figure to include mesh models throughout the PCA axes and 3d projections of the preoptic area’s image with the landmarks. This was repeated for all shape regions and added as Figure Supplements to Figure 5 (Figure 5 —figure supplement 1-6).

I do not see if the original image data and other related data will be shared or not.

There is a data sharing section that has the dryad DOI address where all scripts, raw data, analyzed data and tables can be found. We have also uploaded all code used throughout for the reviewers and readers.

Reviewer #3 (Recommendations for the authors):I thoroughly enjoyed this manuscript, I think it can eventually be a fantastic paper! I would be very happy to look at it again and I have no problems with the data. Please understand that my criticism is basically saying that I don't think you completely know HOW interesting/impactful your results are because you do not address to the long arguments and disagreements in the literature. I think you should spend a while really dissecting what people have been thinking about the evolution of vertebrate brain. Read the vast body of literature on brain evolution, the works of William Hodos, Ann Buttler, Georg Strieder and Glenn Northcutt especially come to mind right away. Your results based on their original work can be that much more insightful. These established authors have raised many questions which you can chime in with your new techniques. It will be a wasted opportunity to just argue that astyanax is a good model, show the reader WHY it is a good model, what do the results mean. I think then it will elevate the paper to an eLife level.I am including one specific set of points on the abstract that illustrate the same issues again and again in the entire text (I'll be blunt because it is easier, keep in mind that I am a fan of the work):

We agree with the reviewer that an important discussion is needed on current unresolved topics underlying evolution of the brain and how the cavefish fills a need to study the mechanisms underlying anatomical and functional evolution of the brain. We have since edited the introduction significantly (especially lines 79-90) that provides background on how the brain is thought to evolve anatomically, along with two paragraphs in the discussion , (1) briefly discussing how unresolved questions can be addressed going forward utilizing non-traditional methods for neurological inquiry, such as the fossil record (e.g. soft tissue is poorly preserved) (line 330-346), and (2) recent studies in other non-model fish species that are providing a way forward for understanding and testing the functional significance of convergent changes in neuroanatomical evolution (lines 348-365).

"Brain anatomy is highly variable and it is widely accepted that anatomical variation impacts brain function and ultimately behavior."I think the initial statement of the abstract is already confounding. You are already not bringing evolutionary neuroscientists to your side. It is too simplistic. I think anyone who knows anything about the issue will either dismiss you, thinking that you have not done your homework in terms the literature, or that your knowledge of the subject is not sophisticated. There is an enormous body of evidence on the literature that show how conserved brains actually are. Diversity and homology are actually one of the main tenants of MANY brain evolutionary studies. Also revise the sentence to avoid repetition of "anatomy/anatomical" for stylistic reasons.

We agree with the reviewer that the vertebrate brain, or bauplan of the brain, has remained remarkably conserved. However, variation in the size, shape and function of individual brain regions can be highly variable and contributes greatly to the amazing diversity of behaviors we see across taxa. We have edited the text in the abstract and introduction to help clarify this point (lines 12-19).

"The structural complexity of the brain, including differences in volume and shape, presents an enormous barrier to define how variability underlies differences in function."I respectfully disagree, the challenge is no longer in this level of analysis. Neurons themselves, circuits and modulation are the next frontier that will inform our questions. I think the first 2 sentences are weak and do not buttress the argument of why this study is important and novel. I would recommend to rewrite it.

We agree with the reviewer that the function and modulation of circuits is fundamental to understanding behavioral evolution. We have edited the text to state that anatomy and function together create obstacles for determining how evolution of the brain results in novel behaviors. We have since rewritten the abstract and focus on how anatomical variation can lead to functional and ultimately behavioral variation (see lines 12-15 and 35-40)

"In this study, we sought to investigate the evolution of brain anatomy in relation to brain region volume and shape across the brain of a single species with variable genetic and anatomical morphs."Again this has been done for years, at this point of reading, I am not convinced why this is novel enough to give insights to the audience of eLife. I think here you really have the chance to drive your point that your results are really important, but you don't yet have the knowledgeable reader on your side.

We agree with the reviewer that our arguments and language did not convey to the reader how the unique characteristics of this model, a single species with variable genetic and anatomical morphs, were leveraged through hybridization to functionally test how every brain region anatomically relates to one-another. We have since changethe sentence to read, “In this study, we sought to investigate the evolution of brain anatomy using a single species of fish consisting of divergent surface and cave morphs, that permits functional genetic testing of regional volume and shape across the entire brain.”(line 17-19) In addition, we believe that added paragraphs addressing previous concerns from the introduction and discussion will help the reader grasp the importance of this model in addressing how anatomical change impacts organismal function and behavior.

[Editors’ note: what follows is the authors’ response to the second round of review.]

The manuscript has been improved but there are some remaining issues that need to be addressed, as outlined below:1. The authors must check whether the eye-size is a confounding factor of the brain region volume changes. This could be done by a correlation analysis of eye-size vs brain regions/shape (PCA1 or PCA2 axis value).

We address this comment below. This is a thoughtful comment and an experiments worth doing, but we did not capture images of the eye. Moreover, because the eyes were not the focus of our imaging, in many samples the eyes extended beyond the focal plane. In other words, the full eyes were not captured in our confocal images, precluding any correlation. This however has been the focus of a collaborative study with the Kowalko lab. While these details are complex for this rebuttal, it is noteworthy that not all regions of the brain correlate with eyes.

We agree this is an important point and have highlighted this in the discussion.

2. The discussion and references on brain evolution need to be significantly expanded.

Thank you. We have now added paragraphs to the discussion. Specifically, we have commented on brain development and how our work fits into this large field; we comment on the potential relationship with eyes; and we comment on how expanded brain regions evolve.

Reviewer #1 (Recommendations for the authors):Thank you to the authors for their thoughtful responses to my concerns. I find the manuscript to be significantly improved. I feel that the arguments they make are now well-supported. I have a few relatively minor comments on places where the writing was not clear.Is the segmentation by tERK with labels for region-specific markers done with ONLY the tERK information? Is there any chance that signal from the regional marker could influence the segmentation?

No, the ANTs registration to provide the reformatting instructions for overlaying a subject brain to the atlas brain is only using the tERK stained image. Once finished, the registration code produces three files that are used to inverse register the segmented atlas file back onto the subject brain. Therefore, the regional marker files are never part of the processes.

Lines 126-132. The figure references do not refer to the correct panels (wrong genes). Also, anti-TH is not present in any figure; please add this.

*We apologize to the reviewer and have updated the text and figure legend.* We have added nitrous oxide 1 and the missing anti-tyrosine hydroxylase image to the multi-panel figure. The text now reads,

Line 126- 132, “We then tested the accuracy of smaller regions, such as the dorsal subpallium, medial preoptic region, and thalamus, via gastrulation brain homeobox 1 (gbx1), oxytocin (oxt) and nitrous oxide 1 (nos1) RNA labeling, respectively (Figure 1 – Supplement 3b-d, [35,36,42,43]). Finally, we confirmed the accuracy of the smallest sub-regions of the brain that can be defined molecularly, such as the locus coeruleus and dorsal raphe, using tyrosine hydroxylase (TH) and 5-hydroxytryptamine (5-HT) antibody labeling, respectively (Figure 1 – Supplement 3eandf, [33,35,44-47]).”

Reviewer #2 (Recommendations for the authors):This is a significant and excellent update to the original submission by Kozol et al. Most of the reviewers' concerns are resolved with a much improved clearer logical flow and newly added Molino cavefish and F2 hybrid data, further adding detailed shape and volume analyses. There are one major concern and several minor points in this resubmitted version of their manuscript.Although the volume change analyses were now more elaborated, it is unclear if the eye size is a correlated factor to explain the antagonistic volume-changes between the dorso-caudal and ventro-rostral brain regions. The authors explained that this pattern of volume change is the result of developmental/genetic constrain with new F2 data, which is convincing. Yet, the major sensory difference at 6 days old – functional vs non-functional eyes (surface fish and cavefish, respectively) – could affect the brain anatomical changes. I assume that authors took whole body pictures before dissection (or, eyes were scanned together during brain imaging) and the eye-size parameter is available to test whether the whole brain landscape is basically correlated with the eye size. None of the other sensors including the olfactory epithelium, lateral line, and inner ear, was reported with significant differences between surface fish and cavefish, as far as I know. I believe the eye size is the important factor in arguing the brain architecture while I was reading through this manuscript, I assumed the brain developmental genes (otx, islet, emx) might regulate these changes but eye developmental genes (pax 6, lens genes) could be the major driver of these brain morphological changes, instead. To answer their research question, whether the whole brain regions or individual brain regions evolved, I believe the eye is the important brain region the authors should consider. F2 hybrids would be a great resource to check it.

We thank the reviewer for this insightful comment. Unfortunately, we did not take images of the eyes before fixing and imaging. Moreover, analyzing eye size from confocal images presented challenges in itself: Because the eyes were not the focus of the image, it is difficult to say where the eye starts and where it ends. Moreover, because the eyes protrude well beyond the brain, we often did not image the entire eye as they were out of the x-y focal range. We feel that making conclusions from these data would be inconstant.

We have begun to address this in a subsequent collaborative study with the Kowalko lab, which we are hoping to put on a pre-print server soon. The focus of this subsequent study is on the relationship between eyes and different regions of the brain. While this is a large study and the findings are out of the scope of this manuscript, it is noteworthy that not all regions that change correlate with eyes, suggesting more complex underlying mechanisms.